# RAG-FGO: Enhancing RAG with Fungal Growth Optimizer for LLM Agents

## Abstract

Generative retrieval leverages large language models (LLMs) to directly generate retrieval queries or candidate document representations, and has recently shown great potential in open-domain question answering and knowledge-intensive tasks. Compared to traditional index-based retrieval methods, generative retrieval provides greater flexibility in handling semantic diversity and complex task requirements. However, existing approaches often rely on static prompt designs or fixed generation strategies, which struggle to maintain both stable and efficient performance in scenarios characterized by semantic complexity, task variability, or noisy knowledge bases. To address these limitations, we propose RAG-FGO (Retrieval-Augmented Generation with Fungal Growth Optimizer), a heuristic search-based framework for optimizing dynamic generative retrieval agents in Retrieval-Augmented Generation (RAG) systems. RAG-FGO integrates both global and local search strategies within the solution space to generate more robust and effective retrieval prompts and parameters, while avoiding local optima. In addition, it introduces a query memory pool that stores high-performing prompts during iterative optimization, thereby guiding subsequent search and generation. Experimental results indicate that RAG-FGO outperforms baselines such as Direct, ReAct, and Self-Act on benchmark datasets including HotpotQA, MuSiQue, and SQuAD, confirming its effectiveness for complex generative retrieval tasks.

## 1 Introduction

Retrieval-Augmented Generation (RAG) (Lewis et al., 2020) has recently been widely applied to knowledge-intensive tasks such as open-domain question answering, fact verification (Su et al., 2024), and domain-specific text generation (Gao et al., 2023). Unlike traditional language models that rely solely on internalized knowledge stored in model parameters (Xu et al., 2024), RAG systems leverage external knowledge bases to dynamically supplement information (Gupta et al., 2024b), thereby improving the accuracy and relevance of generated content (Su et al., 2024).

Within this paradigm, the performance of the retrieval module is a critical determinant of overall system effectiveness (Ye et al., 2024). In particular, the design and construction of retrieval prompts directly influence the quality and utility of the retrieved context (Gao et al., 2023). To enhance retrieval effectiveness, researchers have proposed a range of prompt optimization techniques, primarily including generative retrieval and prompt tuning. The former rewrites original queries using language models to make them semantically clearer and structurally more compatible (Gupta et al., 2024a), thus improving recall and semantic alignment (Lewis et al., 2020). The latter modifies the structure of prompts through methods such as gradient-based optimization or prefix tuning, aiming to better adapt to the task distribution.

Although these approaches have achieved notable success in specific scenarios, they also exhibit clear limitations. Generative query methods often rely on large-scale training data, incur high computational costs, and exhibit limited generalization, making them less adaptable to rapidly changing task requirements. Prompt tuning methods, on the other hand, typically depend on manually designed templates or statically learned structures from specific tasks, lacking the ability to dynamically adjust prompt content in response to new inputs or domains, which often results in rigid retrieval strategies and limited generalization across diverse tasks (Wu et al., 2024; Amatriain, 2024).

Based on these observations, we propose a different perspective: rather than treating generative retrieval as a one-shot query-rewriting task, we model the process of transforming a user's original question into a more complete and retrieval-oriented expression as an optimization-based retrieval agent. During the optimization stage, this agent functions as a parameterized query-rewriting configuration that takes the user question as input, produces a reconstructed retrieval query, and is iteratively selected and improved based on the quality of the retrieved evidence.

To operationalize this agent-based perspective, we build on the Fungal Growth Optimization (FGO) algorithm (Abdel-Basset et al., 2025) and introduce RAG-FGO, a novel framework for dynamic optimization of retrieval agents. Retrieval-Augmented Generation with Fungal Growth Optimizer (RAG-FGO), for dynamic optimization of retrieval agents. Inspired by the adaptive growth behavior of fungal mycelium, RAG-FGO formulates retrieval optimization as a semantic-space search problem, combining global exploration with local refinement to dynamically discover high-quality agents. Furthermore, we introduce a query memory pool to retain top-performing agents, providing stability and guidance for subsequent searches.

Specifically, RAG-FGO begins by initializing a set of semantically diverse candidate retrieval agents, where each agent consists of a query prompt and a parameter vector. These agents are iteratively optimized in the semantic space. In each iteration, the system evaluates agents based on similarity scoring against the target context, selects the best-performing agents as search centers, and applies both global and local search strategies. Local search introduces fine-grained semantic transformations, while global exploration leverages the generative capabilities of large language models to diversify query expressions. To further enhance robustness and cumulative effectiveness, RAG-FGO maintains a query memory pool that continuously preserves the best-performing agents from each iteration, ensuring the long-term accumulation of effective strategies. The final output is the optimal agent identified in the solution space.

We evaluate RAG-FGO on QA benchmarks such as HotpotQA and MuSiQue, and reasoning benchmarks exemplified by MMLU-Pro, comparing against representative baselines including Direct and Zero-shot CoT. RAG-FGO consistently achieves higher accuracy, with notable gains on semantically complex queries requiring multi-hop reasoning and fine-grained contextual alignment. These results demonstrate the robustness of our framework and its effectiveness in overcoming the limitations of existing retrieval optimization methods. Our main contributions are summarized as follows:

- We introduce RAG-FGO, a novel framework that incorporates fungal growth optimization into retrieval-augmented generation, formulating retrieval optimization as a dynamic search problem in semantic space.

- We design an agent-based optimization strategy, where each agent consists of a prompt and parameter configuration. By integrating global exploration, local refinement, and a query memory pool, the framework achieves efficient and stable optimization.

- We conduct extensive experiments on multiple knowledge-intensive QA datasets. Results suggest that RAG-FGO provides reliable gains on complex semantic retrieval tasks relative to existing approaches.

## 2 RELATED WORK

### 2.1 RAG IN LARGE LANGUAGE MODELS

Retrieval-Augmented Generation (RAG) (Lewis et al., 2020) has become a widely adopted solution to the knowledge limitations of LLMs by enriching model parameters with external evidence. Prior work has substantially advanced RAG along two main directions: improving retrieval quality—through dense retrievers (Karpukhin et al., 2020), late-interaction architectures such as Col-BERT (Khattab & Zaharia, 2020), and more scalable retrieval infrastructures—and enhancing the fusion stage through encoder–decoder architectures and retrieval-aware generation modules (Izacard & Grave, 2020; Mialon et al., 2023). Beyond these components, recent research explores multi-stage retrieval pipelines (Mao et al., 2021), reinforcement-learning–driven adaptive retrieval policies (Jiang et al., 2023; Shi et al., 2023), as well as long-context reasoning capabilities that allow models to process substantially larger evidence windows (Bai et al., 2023).

Despite this progress, most RAG pipelines still rely on *static* user queries and lack mechanisms for dynamically adjusting retrieval behavior across diverse tasks, query types, or domains. Static prompts frequently lead to redundant retrieval or mismatched evidence (Gao et al., 2023), limiting the system's ability to adapt to semantic variability. Our work addresses this core limitation by treating query optimization as a learnable retrieval-agent search problem rather than a fixed template or prompt-engineering task.

## 2.2 HEURISTIC SEARCH ALGORITHMS

Heuristic and bio-inspired search algorithms provide effective strategies for high-dimensional, non-convex optimization problems (Talbi, 2009; Blum & Roli, 2003). With the growing reliance on black-box large language models, such methods have been increasingly used for LLM-related optimization tasks where gradients are unavailable or unstable. Prior work explores prompt evolution using genetic algorithms, evolutionary strategies, or population-based search (Fernando et al., 2023; Li et al., 2024; Yang et al., 2023), gradient-free prompt selection (Shin et al., 2020; Prasad et al., 2023), and reinforcement-learning-based prompt tuning (Deng et al., 2022). Other studies employ swarm-intelligence algorithms such as PSO or GA for tuning LLM decoding parameters or steering generation behaviors (Liu et al., 2024; Qolomany et al., 2017).

Bio-inspired methods are particularly compelling for LLM optimization because they naturally balance broad exploration with fine-grained local refinement (Yang, 2020). Among them, the Fungal Growth Optimizer (FGO) (Abdel-Basset et al., 2025) models hyphal expansion and branching to achieve coordinated global–local search. This property aligns well with retrieval optimization, which requires both large semantic jumps and controlled prompt-level adjustments. Building on these strengths, we adapt FGO to the RAG setting and develop a dynamic retrieval-agent optimization framework that iteratively improves query reformulations through semantic-space search.

## 3 METHOD

The core of this study is to construct a generative retrieval agent that identifies optimal prompts and model parameters from a large search space on domain-specific datasets. To this end, we draw on the idea of FGO, inspired by fungal hyphae growth, which emphasizes balancing global exploration and local exploitation to improve optimization efficiency. By iteratively updating candidate prompts and parameters, this idea guides the agent to continuously improve. As shown in Figure 1, the process explores the solution space to refine retrieval. Building on this, we propose the RAG-FGO framework and describe its training workflow and key components in detail.

### 3.1 FGO-BASED GENERATIVE RETRIEVAL

**Task Definition.** We study the task of generative retrieval optimization in a RAG setting. Given a user query $q$ and a knowledge base $\mathcal{K}$, the goal is to generate improved query reformulations that lead to more semantically relevant retrieved contexts. Unlike conventional RAG systems that directly use the raw query or apply a single-step rewrite, we model query optimization as learning a retrieval agent $A = (C, \theta)$, which produces both a prompt template and decoding parameters for generating rewritten queries. The quality of an agent is evaluated by the semantic similarity between the retrieved passages and the reference context. The objective is to identify the agent that yields the highest retrieval quality.

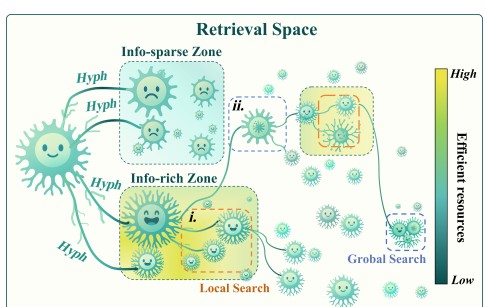

Figure 1: Fungal hyphal growth.

**Overview of FGO Framework.** Figure 2 illustrates the key definitions and operational concepts of the proposed RAG-FGO framework within a generative retrieval system. The framework is designed for dynamic optimization of retrieval prompts. It starts from a natural language query, which is processed by a template initialization module to generate a diverse set of prompt–parameter pairs.

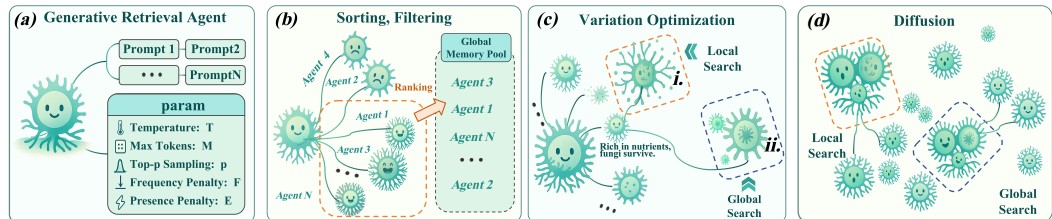

Figure 2: Conceptual illustration of the RAG-FGO framework. (a) A generative retrieval agent generates multiple prompts under different parameter settings. (b) Candidate hypotheses are ranked, and high-quality ones are stored in the global memory pool. (c) Variation optimization integrates local and global search to refine and diversify candidates. (d) The diffusion process propagates promising candidates through iterative search to improve robustness.

Each agent in this set consists of a query template and a set of decoding control parameters (e.g., temperature, top-$p$, and maximum generation length), forming a candidate pool with different query styles and semantic biases. These candidates are then used to retrieve contextual information from an external knowledge base. The retrieved results are compared against reference contexts to compute the retrieval quality of each prompt–parameter pair based on similarity scores. Candidates are ranked accordingly, and high-quality agents are selected.

The top-performing prompt–parameter pairs are stored in a memory pool to preserve historically effective strategies and serve as the starting point for the next iteration of the search process. In each iteration, the framework applies local and global search in parallel, where the local search introduces fine-grained semantic variations (e.g., synonym replacement, tone adjustment) and the global search restructures prompts via LLMs to enable larger semantic shifts, thereby expanding the search space.

Newly generated prompt–parameter pairs are fed into subsequent iterations, with the memory pool continuously updated to accumulate improved strategies. Ultimately, the agent that achieves the best retrieval performance from the memory pool is selected as the final output. Through iterative search and memory-enhanced optimization, RAG-FGO achieves efficient and stable prompt refinement, significantly improving semantic alignment and contextual retrieval quality. The detailed training workflow of RAG-FGO is illustrated in Figure 3.

**Agent initialization.** RAG-FGO first initializes a set of $N$ agents $\mathcal{A} = \{A_1, A_2, \ldots, A_N\}$, where each agent $A_j = (C_j, \theta_j)$ consists of a prompt $C_j$ and parameters $\theta_j \in \mathbb{R}^d$. These agents encode diverse query augmentation strategies, covering multiple intent expressions and prompting styles to meet heterogeneous retrieval needs. Initial prompts are sampled from a predefined prompt space, which may include templates such as: *"As a linguistic diversity expert, you excel at subtly rewriting sentences while preserving their core intent and information."* In parallel, each agent is assigned a decoding parameter vector $\theta_j$ that controls generation behavior, for example: temperature = 0.7, top-$p$ = 0.8, frequency penalty = 2.0, presence penalty = 2.0, max tokens = 1024. Formally, prompts are drawn from a structured template pool, while parameters $\theta_j$ are initialized from a normal distribution to ensure diversity across the agent population:

$$\theta_j \sim \mathcal{N}(\mu_0, \sigma_0^2) \tag{1}$$

where $\mu_0$ and $\sigma_0$ are predefined mean and variance, respectively. This initialization ensures diversity at the starting point of optimization.

**Scoring and Selection.** During training, we first employ the agents selected in the current iteration to generate candidate context passages, which are then compared against the corresponding reference context $c_i^*$ for semantic alignment. Specifically, The semantic similarity between a candidate context $c_{i,j}$ and its reference $c_i^*$ is then computed by measuring the cosine similarity between their corresponding embeddings, reflecting how closely the candidate aligns with the reference in terms of meaning:

$$s_{i,j} = \text{Sim}(c_{i,j}, c_i^*) = \frac{c_{i,j} \cdot c_i^*}{\|c_{i,j}\| \, \|c_i^*\|} \tag{2}$$

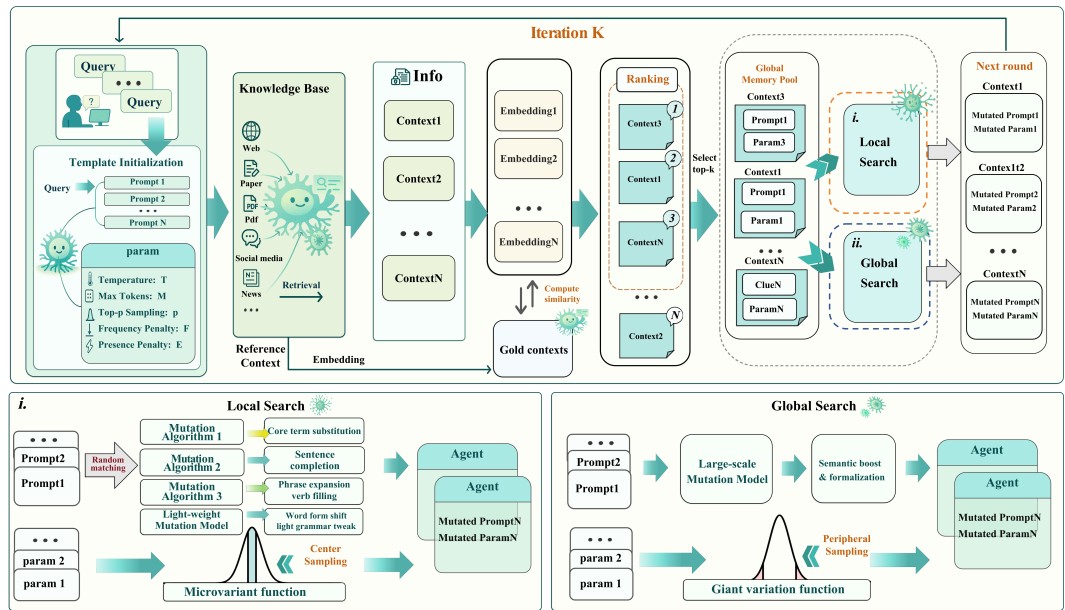

Figure 3: Training process for optimizing a retrieval agent driven by RAG-FGO. Queries are expanded into prompt–parameter templates, ranked by relevance, and evolved through local and global search. The system adaptively explores the retrieval space, guiding the agent toward info-rich zones while diffusing and optimizing query candidates across multiple search rounds.

Here, $c_{i,j}$ and $c_i^*$ denote dense vector representations obtained from a pretrained embedding model, which projects semantic information into a shared vector space. The resulting score $s_{i,j}$ provides a normalized measure of semantic alignment, independent of vector scale.

Based on these scores, all agents are ranked, and the top-$k$ agents are selected according to a maximum selection rule:

$$\mathcal{A}_{\text{top-}k} = \text{Top-k}\big(\{(A_j, s_{i,j}) \mid A_j \in \mathcal{A}\}\big) \tag{3}$$

This mechanism ensures that, in each iteration, only the prompt–parameter pairs producing candidate outputs most aligned with the reference contexts are retained. The selected agents are subsequently stored in the query memory pool $\mathcal{M}$, which acts as a global repository of high-quality candidates. By retaining the best-performing agents across iterations, the memory pool facilitates cumulative optimization, mitigates catastrophic forgetting, and provides a stable foundation for subsequent variation and diffusion processes.

**Exploration and Exploitation.** To promote agent diversity and prevent the search process from becoming trapped in local optima, RAG-FGO uses both local search and global exploration for each selected agent $A_j = (C_j', \theta_j)$. At the beginning of each outer iteration $t$, a dynamic local search probability $p_t$ is scheduled between predefined minimum and maximum values $p_{\min}$ and $p_{\max}$:

$$p_t = p_{\min} + \frac{t-1}{T_{\max} - 1}(p_{\max} - p_{\min}) \tag{4}$$

This scheduling strategy ensures that early iterations emphasize global exploration ($p_t$ small), while later iterations gradually shift toward local refinement ($p_t$ large). For each mutation, the algorithm samples from $\text{Bernoulli}(p_t)$ to decide whether to perform local search or global exploration. Local search introduces slight semantic modifications to the prompt and applies small perturbations to the parameters:

$$C_j'' = m_{\text{local}}(C_j', \epsilon_c^{(l)}), \quad \theta_j' = \theta_j + \epsilon_\theta^{(l)}, \quad \epsilon_\theta^{(l)} \sim \text{TruncNorm}(0, \sigma_l^2, [-k_l\sigma_l, k_l\sigma_l]) \tag{5}$$

Here, $\text{TruncNorm}(0, \sigma_l^2, [-k_l\sigma_l, k_l\sigma_l])$ denotes a truncated normal distribution with mean 0 and variance $\sigma_l^2$, truncated to $[-k_l\sigma_l, k_l\sigma_l]$. The hyperparameter $k_l$ constrains the perturbation range so that sampled values are concentrated near zero.

In contrast, global exploration introduces more substantial modifications to both the prompt and the parameters. It excludes perturbations near zero by sampling only values whose absolute magnitude exceeds a threshold $\delta$:

$$C''_j = m_{\text{global}}(C'_j, \epsilon_c^{(g)}), \quad \theta'_j = \theta_j + \epsilon_\theta^{(g)}, \quad \epsilon_\theta^{(g)} \sim \text{TruncNorm}(0, \sigma_g^2, [-k_g\sigma_g, -\delta] \cup [\delta, k_g\sigma_g]) \quad (6)$$

where $\delta > 0$ specifies the minimum allowable perturbation magnitude and $k_g$ sets the maximum range.

**Query Memory Pool.** The query memory pool $\mathcal{M}$ serves as a global repository for retaining high-quality agents across optimization iterations. Formally, it maintains the set of agent–score pairs:

$$\mathcal{M} = \{(A_j, s_{i,j})\}_{j=1}^M \quad (7)$$

In each iteration, the top-$k$ agents according to their scores $s_{i,j}$ are inserted into $\mathcal{M}$. To prevent uncontrolled growth, older or lower-scoring agents may be replaced, ensuring that the memory pool reflects both recent updates and historically strong candidates. This design enables the long-term accumulation of diverse, high-quality prompts while avoiding catastrophic forgetting.

By integrating information across iterations, the memory pool provides a stable foundation for subsequent variation and diffusion steps, allowing RAG-FGO to refine its search space without repeatedly exploring low-quality regions. At the end of training, the agent with the highest score stored in $\mathcal{M}$ is selected as the final output and deployed as the generative retrieval agent during testing. This mechanism ensures that the final model inherits both the stability of cumulative optimization and the efficiency of memory-enhanced selection.

**Output and Application.** Upon completion of training, RAG-FGO selects the highest-scoring retrieval agent from the memory pool $\mathcal{M}$:

$$A^* = \arg \max_{A_j \in \mathcal{M}} s_{i,j} \quad (8)$$

The selected agent $A^* = (C^*, \theta^*)$ contains both the optimized prompt $C^*$ and the optimized parameters $\theta^*$. This agent is then deployed in the testing phase for retrieval and context selection. This ensures that the evaluation is conducted using the most effective retrieval strategy discovered during training. The pseudocode of the overall framework is provided in Appendix 1.

## 4 EXPERIMENTS

### 4.1 EXPERIMENTAL SETUP

**Datasets.** We evaluate our methods on two groups of benchmarks. (1) *QA benchmarks:* Long-Bench (Bai et al., 2023) targets long-context understanding with inputs up to 100 K tokens. HotpotQA (Yang et al., 2018) is a multi-hop QA dataset over multiple supporting documents. 2WikiMultihopQA (Ho et al., 2020) focuses on compositional multi-hop questions in Wikipedia. MuSiQue (Trivedi et al., 2022) decomposes multi-hop reasoning into single-hop questions. Natural Questions (NQ) (Kwiatkowski et al., 2019) contains real queries with Wikipedia passages. SQuAD (Rajpurkar et al., 2016) is a reading comprehension dataset with span-based answers. TriviaQA (Joshi et al., 2017) is a large-scale open-domain QA dataset with web and Wikipedia evidence.

(2) *Reasoning benchmarks:* MATH500 (Hendrycks et al., 2021) is a subset of the MATH dataset containing 500 challenging competition-level problems. MMLU-pro (Wang et al., 2024) extends the standard MMLU benchmark with more professional and domain-specific tasks. GPQA (Rein et al., 2024) evaluates graduate-level problem-solving ability in science and engineering domains. DROP (Dua et al., 2019) is a reading comprehension dataset that requires discrete reasoning such as addition, counting, and comparison. MuSR (Sprague et al., 2023) is a benchmark for multi-step symbolic reasoning, containing two subsets: MuSR-op and MuSR-ta.

**Baselines.** We compare RAG-FGO with two groups of representative methods. (1) *Retrieval-based baselines:* Direct lets the LLM answer questions without retrieval. ReAct (Yao et al., 2023) integrates CoT reasoning with retrieval tool use. Self-Act (Press et al., 2022) (Self-Ask with Search) decomposes a query into sub-questions and retrieves evidence for each. Aflow (Zhang et al., 2024a) employs an adaptive flow strategy that dynamically decides when to reason or retrieve, making it effective for multi-hop QA. (2) *Reasoning-based baselines:* Zero-shot CoT elicits reasoning by adding

a trigger phrase (e.g., "Let us reason step by step") to the query without in-context exemplars. Few-shot CoT guides reasoning by providing a small set of annotated exemplars, allowing the model to learn from in-context demonstrations. USC (Chen et al., 2023) generates multiple CoT reasoning paths and selects the final answer by majority voting. CoT-WP (Wang & Zhou, 2024) weights reasoning paths by confidence scores from token-level probabilities and aggregates them via weighted voting. LEAP (Zhang et al., 2024b) enhances few-shot prompting by inducing errors on exemplars and using self-reflection to extract transferable reasoning principles. ReFeri (Lee et al., 2025a) recycles few-shot exemplars to score and verify candidate outputs, selecting responses that are both confident and contextually coherent without additional training.

**LLM Models.** We do experiments with a range of proprietary and open-source LLMs: GPT-4o-mini and GPT-4.1-nano, representing compact yet strong proprietary models; Qwen-Long and Qwen-Max (Qwen team), designed for long-context reasoning and high performance; and Gemma2-27B (Google DeepMind), a large open-source model optimized for efficiency and accuracy.

**Training vs. Inference.** Our framework aims to train a dedicated retrieval agent. During **training**, we rely on supervision signals (e.g., gold supporting passages and annotated answers) to evaluate retrieval quality and guide the optimization of the agent via RAG-FGO. These references are used solely to shape the optimization landscape and are not required at test time. Once training converges, the agent $\pi_{\phi^\star}$ directly maps any new query $q$ to an optimized prompt–parameter pair $(C^\star, \theta^\star)$ in a single forward pass. In **inference**, RAG uses only $(q, C^\star, \theta^\star)$ for retrieval and generation, without any access to gold contexts or labels. Specifically, given a new query $q$, the optimized prompt $C^\star$ together with the tuned parameters $\theta^\star$ are employed to retrieve relevant passages from the external knowledge source. These retrieved passages are then supplied to the generator, which produces the final answer conditioned solely on $(q, C^\star, \theta^\star)$ and the retrieved evidence. This ensures that the agent operates efficiently and autonomously at test time, relying entirely on its learned retrieval strategy rather than external supervision.

For the main experiments, we adopt fixed configuration combinations for each dataset. On GPT-4o-mini, we set iter=4, mc=9 for LongBench and iter=5, mc=12 for HotpotQA, where mc denotes the per-iteration search count (i.e., the number of candidate prompt–parameter variations explored in each round). Similar task-specific configurations are applied across other datasets to balance accuracy and efficiency. These combinations serve as the standard settings for the reported results in Table 1, with optimal iteration–configuration combinations further analyzed in the ablation study.

Table 1: Performance of search strategies (Direct, ReAct, Self-Act, Aflow, and RAG-FGO) across six QA benchmarks. All metrics are reported as F1 scores (%).

| Model | Method | LongBench | HotpotQA | 2WikiMultihopQA | MuSiQue | NQ | SQuAD | AVG |
|---|---|---|---|---|---|---|---|---|
| GPT-4o-mini | Direct | 58.1 | 51.1 | 48.9 | 40.1 | 36.4 | 51.2 | 47.6 |
| | ReAct | 62.7 | 65.6 | 52.3 | 53.6 | 48.9 | 55.3 | 56.4 |
| | Self-Act | 68.9 | 73.1 | 69.2 | 67.3 | 62.4 | 69.5 | 68.4 |
| | Aflow | 61.0 | 73.5 | 65.7 | 71.6 | 67.9 | 73.7 | 68.9 |
| | **RAG-FGO** | **71.5** | **81.5** | **77.3** | **79.4** | **73.8** | **79.2** | **77.1** |
| GPT-4.1-nano | Direct | 56.3 | 48.5 | 45.3 | 38.2 | 33.7 | 49.3 | 45.2 |
| | ReAct | 59.7 | 61.3 | 48.9 | 47.3 | 45.4 | 50.9 | 52.2 |
| | Self-Act | 64.5 | 69.7 | 62.2 | 61.1 | 59.8 | 62.7 | 63.3 |
| | Aflow | 66.1 | 74.2 | 70.1 | 69.4 | 66.5 | 72.1 | 69.7 |
| | **RAG-FGO** | **67.8** | **77.8** | **75.7** | **73.1** | **70.3** | **77.6** | **73.7** |
| Qwen-Long | Direct | 59.3 | 55.7 | 51.9 | 55.4 | 43.4 | 54.3 | 53.3 |
| | ReAct | 64.2 | 69.2 | 57.2 | 58.9 | 56.3 | 58.7 | 60.8 |
| | Self-Act | 68.3 | 76.4 | 73.1 | 72.3 | 68.9 | 70.4 | 71.6 |
| | Aflow | 70.8 | 80.1 | 76.8 | 78.9 | 74.6 | 77.1 | 76.4 |
| | **RAG-FGO** | **72.3** | **84.3** | **80.2** | **82.6** | **79.1** | **80.4** | **79.8** |
| Qwen-Max | Direct | 47.2 | 52.7 | 46.6 | 49.9 | 53.3 | 48.5 | 49.7 |
| | ReAct | 52.3 | 60.1 | 53.1 | 53.6 | 59.8 | 54.5 | 55.6 |
| | Self-Act | 58.7 | 65.6 | 64.8 | 61.8 | 67.8 | 68.3 | 64.5 |
| | Aflow | 61.2 | 71.4 | 68.1 | 65.3 | 73.9 | 74.0 | 69.0 |
| | **RAG-FGO** | **63.9** | **76.6** | **71.1** | **67.4** | **77.3** | **78.2** | **72.4** |
| Gemma-2-27B | Direct | 49.3 | 49.7 | 43.4 | 48.4 | 51.3 | 48.5 | 48.4 |
| | ReAct | 52.5 | 56.3 | 50.8 | 51.7 | 57.8 | 54.5 | 53.9 |
| | Self-Act | 56.4 | 60.5 | 59.7 | 59.7 | 65.3 | 65.7 | 61.2 |
| | Aflow | 60.8 | 69.1 | 63.2 | 63.4 | 69.4 | 71.9 | 66.3 |
| | **RAG-FGO** | **64.9** | **74.5** | **66.7** | **66.1** | **73.1** | **76.6** | **70.3** |

## 4.2 PERFORMANCE EVALUATION

**RAG Baselines (retrieval-based).** We comprehensively evaluate RAG-FGO against four representative baselines—Direct, ReAct, Self-Act, and Aflow—on six widely used QA benchmarks, including LongBench, HotpotQA, 2WikiMultihopQA, MuSiQue, NQ, and SQuAD, and report results using the standard answer-level F1 metric to ensure fair and consistent comparison across datasets.

As shown in Table 1, RAG-FGO consistently outperforms all baselines across five model backbones. On GPT-4o-mini, RAG-FGO attains 73.8% on NQ, surpassing Direct at 36.4% by 37.4 points and ReAct at 48.9% by 24.9 points. On SQuAD, it achieves 79.2%, outperforming Direct by 28.0 points and Self-Act by 9.7 points. On HotpotQA, it obtains 81.5%, exceeding ReAct at 65.6% by 15.9 points and Self-Act at 73.1% by 8.4 points. On MuSiQue, it achieves 79.4%, outperforming ReAct at 53.6% by 25.8 points and Self-Act at 67.3% by 12.1 points.

Similar gains are observed on other backbones. For instance, RAG-FGO delivers 84.3% on HotpotQA with Qwen-Long, 76.6% with Qwen-Max, and 74.5% with Gemma-2-27B. Overall, RAG-FGO demonstrates clear superiority over baseline methods, achieving substantial improvements in retrieval accuracy and overall performance across diverse models and datasets.

Table 2 presents a comparison between our RAG-FGO framework and several classical RAG systems, including RAG, FiD, and RE-RAG, on two standard open-domain QA benchmarks: NQ and TriviaQA. To ensure a fair comparison, we constrain the generator size of all methods to $\leq$ 2B and evaluate them under identical retrieval settings, including the same number of candidate documents, unified input formatting, and a consistent evaluation pipeline.

Table 2: EM results on NQ and TriviaQA benchmarks.

| Method | NQ | TriviaQA |
|---|---|---|
| RAG (Lewis et al., 2020) | 44.5 | 56.8 |
| FiD-base (Izacard & Grave, 2021) | 48.2 | 65.0 |
| FiD-large (Izacard & Grave, 2021) | 51.4 | 67.6 |
| RE-RAG-base (Kim & Lee, 2024) | 49.9 | 68.2 |
| RE-RAG-large (Kim & Lee, 2024) | 54.0 | 70.2 |
| **RAG-FGO** | **58.4** | **73.0** |

As shown in the table, RAG-FGO achieves substantially higher EM scores on both datasets. Under the same generator configuration, RAG-FGO consistently outperforms traditional RAG, FiD, and RE-RAG baselines, demonstrating that optimizing the retrieval agent alone yields stable and meaningful performance improvements even without increasing model size. Overall, RAG-FGO enhances open-domain QA performance by optimizing the retrieval agent while keeping the model architecture and retrieval budget unchanged. This indicates that retrieval strategies still offer considerable room for improvement, and RAG-FGO effectively exploits this potential.

Table 3: Comparison of our proposed RAG-FGO with baseline methods across seven reasoning benchmarks using GPT-4o-mini. Best results are highlighted in bold.

| Method | MATH500 (Acc.) | MMLU-pro (Acc.) | GPQA (Acc.) | DROP (EM/F1) | HotpotQA (EM) | MuSR-op (Acc.) | MuSR-ta (Acc.) | Avg. |
|---|---|---|---|---|---|---|---|---|
| Zero-shot CoT | 76.4 | 63.0 | 43.0 | 77.6/85.6 | 31.5 | 58.1 | 56.2 | 58.0 |
| Few-shot CoT | 75.2 | 63.0 | 41.3 | 76.8/83.1 | 34.0 | 59.4 | 77.0 | 61.0 |
| USC | 78.6 | 62.5 | 42.4 | 78.8/85.8 | 36.6 | 59.8 | 76.4 | 62.2 |
| CoT-WP | 77.8 | 64.2 | 42.4 | 77.6/83.1 | 34.0 | 57.0 | 79.6 | 61.8 |
| LEAP | 74.5 | 63.2 | 43.9 | 75.8/83.0 | 34.0 | 59.8 | 74.4 | 60.8 |
| ReFeri | **78.2** | **65.0** | 42.4 | **79.6/85.3** | **36.2** | 61.3 | 82.8 | 63.6 |
| **RAG-FGO** | 78.0 | 64.5 | **44.1** | 79.2/85.0 | 35.9 | **61.7** | **83.1** | **63.7** |

**Reasoning-only Baselines (prompt-based).** On the seven reasoning benchmarks reported in Table 3, RAG-FGO achieves the highest overall average score of 63.7, slightly higher than ReFeri at 63.6. On GPQA, RAG-FGO reaches 44.1%, exceeding ReFeri at 42.4% by 1.7 points. On MuSR-op, it achieves 61.7%, surpassing ReFeri at 61.3% by 0.4 points. On MuSR-ta, RAG-FGO obtains 83.1%, exceeding ReFeri at 82.8% by 0.3 points. These results demonstrate clear advantages in tasks requiring multi-hop reasoning or structured external retrieval. In contrast, ReFeri performs slightly better on several benchmarks. On MATH500, RAG-FGO achieves 78.0%, falling short of ReFeri at 78.2% by 0.2 points. On MMLU-pro, it records 64.5%, lower than ReFeri at 65.0% by 0.5 points. On DROP, RAG-FGO reaches 79.2%/85.0%, underperforming ReFeri at 79.6%/85.3% by 0.4/0.3

points. On HotpotQA, it achieves 35.9%, slightly below ReFeri at 36.2% by 0.3 points, where we adopt the stricter EM metric from the official evaluation script to increase evaluation diversity and ensure consistency with other reasoning benchmarks.

Overall, RAG-FGO demonstrates stable advantages in knowledge-intensive and multi-hop reasoning tasks, while ReFeri maintains marginal superiority in benchmarks that emphasize symbolic computation or rely less on retrieval, with most differences remaining within 0.2 to 0.5 points.

**Efficienc.** Table 4 reports the computational cost of RAG-FGO during the optimization phase on

Table 4: Computation cost of RAG-FGO on HotpotQA: per-iteration cost and full-search cost. All error values are standard deviation from five independent experiments.

| iter | mc | Cost/iter (tokens) | Full cost (tokens) | Cost (full,$) | Runtime (full,s) |
|------|-----|--------------------|--------------------|--------------------|--------------------|
| 3 | 3 | $10,842 \pm 540$ | $32,613 \pm 1,631$ | $0.00489 \pm 0.00024$ | $157 \pm 16$ |
| 3 | 6 | $17,231 \pm 910$ | $51,020 \pm 2,551$ | $0.00765 \pm 0.00038$ | $246 \pm 25$ |
| 4 | 3 | $10,842 \pm 540$ | $40,188 \pm 2,009$ | $0.00603 \pm 0.00030$ | $193 \pm 19$ |
| 4 | 6 | $17,231 \pm 910$ | $69,221 \pm 3,461$ | $0.01038 \pm 0.00052$ | $323 \pm 30$ |

HotpotQA, including the average token consumption per iteration, the total token usage over the full optimization process, the corresponding dollar cost, and the runtime. The results show that both the number of iterations (iter) and the number of candidates per iteration (mc) have a roughly linear effect on the optimization cost: when mc increases from 3 to 6, the per-iteration token usage increases accordingly, and increasing the number of iterations from 3 to 4 under a fixed mc further raises the total cost. Under the highest-budget configuration (4 iterations × 6 candidates), the total token consumption of the optimization process is approximately 69k tokens, with a cost of about $0.01 and a runtime of around 320 seconds. These results indicate that the optimization phase of RAG-FGO remains computationally manageable and exhibits a predictable linear growth trend under different iteration and candidate configurations.

## 4.3 ABLATION STUDY

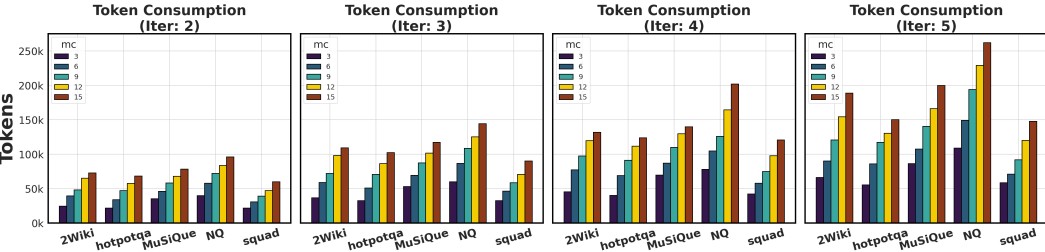

Figure 4: Token consumption across five datasets under different iteration counts (2–5). Each sub-plot corresponds to a fixed iteration count setting, with grouped bars showing mc values (3–15). The y-axis is measured in thousands of tokens (k).

**Impact of RAG-FGO Iterations on Search Performance.** We conduct ablation experiments on the GPT-4o-mini backbone to systematically analyze the impact of iteration count (iter) and search count (mc) on the performance of RAG-FGO, and to further explore how to achieve a reasonable trade-off between accuracy gains and computational cost. The results are shown in Figure 5 and Figure 4. Figure 5 demonstrates that as the number of iterations and searches increases, the model consistently improves across six benchmark datasets, but the gains gradually saturate under higher configurations. For example, on HotpotQA, accuracy increases from 77.6% at iter=2, mc=3 to 81.5% at iter=5, mc=12, while further increasing mc to 15 yields almost no additional improvement. At the same time, Figure 4 shows that token consumption grows nearly linearly with both iter and mc, and becomes significantly amplified at higher iteration counts. On NQ, for instance, the token cost at iter=5, mc=15 is nearly twice that of iter=4,

mc=9, while accuracy improves by only about 1–2 percentage points. Based on the joint analysis of accuracy and cost, we select the most cost-effective configurations for each dataset in the main experiments. For example, we use iter=4, mc=9 on LongBench, achieving 71.5% accuracy while maintaining efficiency; iter=5, mc=12 on HotpotQA, reaching 81.5% accuracy with acceptable cost; iter=4, mc=12 on MuSiQue, obtaining 79.4% accuracy with the best trade-off; iter=4, mc=9 on NQ, achieving 73.8% accuracy without incurring extra overhead; iter=3, mc=9 on 2WikiMultihopQA, where 77.3% accuracy already approaches the optimum; and iter=5, mc=9 on SQuAD, reaching the best accuracy of 79.2%. Overall, these ablation studies confirm not only the effectiveness and stability of RAG-FGO under deeper iterations and searches but also provide rational configurations for different datasets, thereby ensuring the representativeness and reliability of the main experimental results.

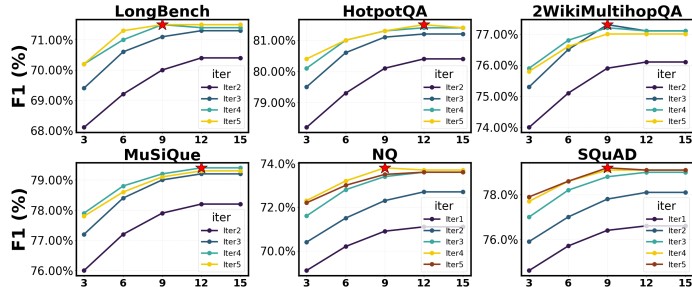

Figure 5: Ablation results of iteration count (iter) and per-epoch search count (mc) on six QA benchmarks with GPT-4o-mini. Accuracy (F1, %) increases with larger mc and iter, but gains saturate at higher values. Red stars mark the chosen configurations that balance accuracy and token cost for the main experiments.

**Component Ablation.** Table 5 reports the ablation results of RAG-FGO across five QA datasets. Removing any component leads to a performance decline, with varying degrees of impact. Removing the memory pool causes the largest drop, with performance falling short of the full model by roughly 2–4 F1 points on average. Eliminating the global search module also lowers performance by around 5–8 points, underscoring its importance for broader exploration. Removing the local search module results in a smaller decline of about 4–6 points, reflecting its role in fine-grained refinement. Overall, these results show that memory, local search, and global search provide complementary benefits, and the full configuration is required for the best performance.

Table 5: Component ablation of RAG-FGO across datasets (GPT-4o-mini). We report answer-level F1 (%), and the relative change w.r.t. the full model.

| Configuration | HotpotQA | MuSiQue | 2Wiki | NQ | SQuAD |
|---|---|---|---|---|---|
| **RAG-FGO (full)** | 81.5 | 79.4 | 77.3 | 73.8 | 79.2 |
| w/o memory pool | 78.3 (-3.2) | 75.4 (-4.0) | 74.0 (-3.3) | 71.2 (-2.6) | 76.1 (-3.1) |
| w/o local search | 75.2 (-6.3) | 72.8 (-6.6) | 70.9 (-6.4) | 66.9 (-6.9) | 72.1 (-7.1) |
| w/o global search | 76.4 (-5.1) | 74.2 (-5.2) | 72.1 (-5.2) | 69.0 (-4.8) | 74.3 (-4.9) |

## 5 CONCLUSION

This paper proposes RAG-FGO, a dynamic optimization framework for Retrieval-Augmented Generation (RAG) systems. Inspired by the adaptive growth of fungal hyphae, RAG-FGO models retrieval optimization as a semantic-space search process, in which agents, comprising both prompts and parameters, are iteratively refined through coordinated local and global search. A query memory pool preserves high-performing agents across successive iterations, ensuring stability and effectively guiding subsequent exploration. We evaluate RAG-FGO on widely used knowledge-intensive QA benchmarks including HotpotQA, MuSiQue, and SQuAD. Experimental results suggest that RAG-FGO achieves steady and consistent improvements, particularly on semantically complex queries, thereby indicating its strong utility for enhancing retrieval quality.In future work, we plan to explore the extension and adaptability of the framework to cross-modal retrieval tasks as well as real-world application scenarios.

# 6 REPRODUCIBILITY STATEMENT

We have taken several steps to ensure the reproducibility of our work. All datasets used in our experiments are publicly available, including HotpotQA, MuSiQue, SQuAD, LongBench, 2WikiMultihopQA, NQ, MATH500, MMLU-pro, GPQA, DROP, and MuSR. Detailed dataset descriptions and references are provided in Section 4.1. To promote transparency, we describe the experimental setup in detail, including backbone models (GPT-4o-mini, GPT-4.1-nano, Qwen-Long, Qwen-Max, and Gemma2-27B), prompt templates, and search hyperparameters such as iteration count and per-iteration search count (iter, mc).

We report all baseline implementations using their official or widely adopted open-source implementations, with consistent evaluation metrics (F1, EM, or accuracy, depending on the benchmark). For fair comparison, we adopt standard preprocessing and evaluation scripts released by the dataset authors. Hyperparameter settings are explicitly stated in the main text and ablation study (Section 4.3), where we analyze the impact of iterations and search counts on both performance and computational cost.

The proposed RAG-FGO framework can be reproduced with access to the same LLM APIs or open-source models and the provided datasets. Pseudocode outlining the full algorithm is included in Appendix A.2. To further facilitate reproducibility, we will release our code, configuration files, and query memory pool implementation upon publication. We will also provide details of computational resources (e.g., GPU type, memory, batch sizes, runtime) to allow researchers to estimate reproduction costs.

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

CONTENTS

# A APPENDIX

## A.1 DETAILED RELATED WORK

### A.1.1 RAG IN LARGE LANGUAGE MODELS

As LLMs continue to scale, the knowledge encoded within their parameters increasingly faces the challenges of *stale updates* and *limited coverage* Roberts et al.. These limitations become particularly pronounced in tasks such as factual question answering, domain-specific knowledge generation, and open-domain dialogue, where relying solely on implicitly stored knowledge often falls short of delivering responses with high accuracy and broad informational coverage. Consequently, recent research has turned towards enhancing the dynamic access of LLMs to external knowledge sources, aiming to improve the factual consistency and information richness of the generated outputs Borgeaud et al. (2022); Li et al. (2022); Pan et al. (2024); Zheng et al. (2025).

RAG is a hybrid generation framework proposed to address this need Lewis et al. (2020); Jiang et al. (2023). The core idea behind RAG is to incorporate a retrieval mechanism into the generation process, enabling the model to access external documents during response generation rather than relying solely on its internal parameters Borgeaud et al. (2022). Concretely, RAG first retrieves the most relevant document fragments from a pre-constructed knowledge base using vector search techniques such as FAISS or Dense Passage Retrieval (DPR) Johnson et al. (2019); Karpukhin et al. (2020). These retrieved fragments are then concatenated with the user query and fed into the generative model, significantly enhancing the model's knowledge coverage and interpretability Izacard & Grave (2020); Guu et al. (2020).

Typically, RAG adopts a dual-encoder architecture, where one encoder models the user query and the other encodes the document collection. This independent encoding setup improves retrieval efficiency and scalability Lewis et al. (2020); Karpukhin et al. (2020); Khattab & Zaharia (2020). In practical implementations, Transformer-based architectures are widely used to integrate the user input and retrieved content, achieving effective synergy between knowledge retrieval and language modeling Vaswani et al. (2017); Mialon et al. (2023).

As the RAG paradigm gains traction, researchers have explored various strategies to further improve its performance. These efforts include: (1) incorporating sparse representation-based retrieval methods such as ColBERT to enhance relevance and inference efficiency Khattab & Zaharia (2020); Santhanam et al. (2021); (2) developing multi-document fusion and knowledge selection mechanisms to mitigate the impact of redundancy and conflicting information Izacard & Grave (2020); Soudani et al. (2024); Zhang et al. (2025); and (3) implementing multi-stage pipelines that follow a retrieve-rerank-generate scheme to improve overall response quality and system robustness Mao et al. (2021); Cheng et al. (2025). More recently, attempts have been made to integrate RAG with reinforcement learning and meta-learning techniques to dynamically adjust retrieval paths, personalize user preferences, and explore a closed-loop "retrieve-control-generate" framework Sutton et al. (1998); Hospedales et al. (2021); Jiang et al. (2023); Shi et al. (2023).

These developments mark a shift from static knowledge generation to knowledge-aware generation and offer more flexible and controllable solutions for complex tasks such as multimodal processing and long-text generation.

However, when dealing with high-dimensional and complex tasks, current approaches often introduce a large amount of redundant information during the document integration phase (e.g., vector recall and fusion ranking), resulting in lengthy retrieved contexts and inefficient computation Lewis et al. (2020); Gao et al. (2023). To address this issue, this paper proposes a Fungal Growth-inspired Optimization strategy for the RAG framework, aiming to suppress redundant information, reduce unnecessary resource consumption, and improve generation efficiency.

### A.1.2 HEURISTIC SEARCH ALGORITHM

In recent years, heuristic search algorithms Pearl (1984); Edelkamp & Schrödl (2011) have demonstrated remarkable performance in solving complex optimization problems, and have been widely applied in key areas such as path planning, resource scheduling, combinatorial optimization, and the optimization of AI models Talbi (2009); Hutter et al. (2019); Bengio et al. (2021). Compared

to traditional deterministic optimization methods, such as gradient descent and linear programming Robbins & Monro (1951); Nocedal & Wright (1999); Ruder (2016); Dantzig (2016), heuristic approaches are better equipped to handle high-dimensional, non-convex, nonlinear, and dynamically changing environments, making them particularly effective for large and structurally complex search spaces Dokeroglu et al. (2019); Yang (2020). These algorithms enhance optimization flexibility and adaptability by incorporating problem-specific heuristics, probabilistic perturbation mechanisms, or collaborative population strategies, thereby improving global exploration capabilities and local convergence efficiency Blum & Roli (2003).

Amid the rapid development of artificial intelligence, especially in large-scale models such as Large Language Models (LLMs) Vaswani et al. (2017); Devlin et al. (2019); Brown et al. (2020), the demands on optimization algorithms have increased significantly. Key tasks in LLMs—such as hyperparameter tuning, neural architecture search, and prompt engineering—are essentially complex high-dimensional search problems Hutter et al. (2019); Brown et al. (2020); Liu et al. (2023). Heuristic search has shown great potential in these domains and is becoming a powerful complement or even an alternative to traditional deep learning optimization methods Elsken et al. (2019); Dokeroglu et al. (2019); Bengio et al. (2021).

Currently, mainstream heuristic search algorithms can be broadly classified into three categories: single-point search based on heuristic functions, population-based parallel search, and bio-inspired optimization strategies Blum & Roli (2003); Talbi (2009). The first category includes classical algorithms such as A* and simulated annealing, which rely on well-designed heuristics or stochastic perturbations and perform effectively in structured problems with clear priors Hart et al. (1968); Kirkpatrick et al. (1983). The second category, grounded in swarm intelligence, includes methods like Genetic Algorithms (GA) and Particle Swarm Optimization (PSO), which utilize information exchange and co-evolution among multiple agents to explore the solution space. These algorithms offer strong global search capabilities and the ability to escape local optima, showing robustness and flexibility in high-dimensional, multimodal settings Mitchell (1998); Qolomany et al. (2017). The third category consists of bio-inspired optimization methods motivated by natural phenomena such as evolution, immunity, and swarm behavior. Representative algorithms include Artificial Immune Systems, Firefly Algorithms, and Ant Colony Optimization De Castro & Timmis (2002); Dorigo et al. (2007); Yang (2009); Dokeroglu et al. (2019), which simulate information exchange and self-organizing behavior among individuals to construct highly adaptive and self-regulating search strategies. These are particularly effective in tackling non-convex, multi-modal, and multi-objective optimization challenges.

In the practical optimization of LLMs, traditional gradient-based methods depend heavily on costly backpropagation and global parameter updates, which are inefficient in dealing with ultra-large parameter spaces. This inefficiency is especially pronounced in tasks such as prompt selection, pruning strategy generation, and model architecture design, where traditional methods often suffer from local optima and low search efficiency LeCun et al. (2015); Han et al. (2015); Bottou et al. (2018).

To overcome these limitations, an increasing number of researchers are integrating heuristic search methods to enhance optimization performance. For instance, some studies have utilized Genetic Algorithms to evolve prompts automatically, significantly improving generalization in few-shot learning and instruction-tuning tasks Liu et al. (2024). Others have applied Particle Swarm Optimization to adjust key architectural parameters such as layer count and hidden dimensions, thereby improving both training efficiency and final performance Qolomany et al. (2017). Notably, bio-inspired optimization methods are gaining traction as an effective strategy for complex and dynamic AI system optimization. Owing to their adaptive search abilities and capacity to manage multi-modal objectives, they offer a robust solution for tackling intricate optimization challenges. Lee et al. (2025b).

## A.2 PSEUDOCODE

We introduce RAG-FGO, a training framework for retrieval-augmented generation that dynamically refines retrieval prompts through iterative search and memory-guided refinement. The algorithm takes as input a training dataset $D$, consisting of sample pairs $(C_i, c_i^*)$, as well as the following parameters: the number of agents $k$, the maximum number of outer iterations $T_{\max}$, the number of inner iterations $S$, the number of searches per agent $n$, and the number of top-performing agents retained per round $r$. At initialization, the framework creates an empty memory pool $\mathcal{M}$ and randomly

---

**Algorithm 1** Training Procedure of RAG-FGO

---

**Require:** Training dataset $D = \{(C_i, c_i^*)\}_{i=1}^N$, number of agents $k$, outer iterations $T_{\max}$, inner iterations $S$, searches per agent $n$, retained agents per round $r$, min/max local prob $p_{\min}, p_{\max}$
**Ensure:** Best agent $A^*$
1: Initialize memory pool $\mathcal{M} \leftarrow \emptyset$, initialize $k$ agents $\mathcal{A}_0 = \{A_j\}_{j=1}^k$
2: **for** $t = 1$ to $T_{\max}$ **do**
3:     **Dynamic schedule:** $p_t \leftarrow \begin{cases} p_{\min}, & T_{\max} = 1 \\ p_{\min} + \dfrac{t-1}{T_{\max}-1}(p_{\max} - p_{\min}), & \text{otherwise} \end{cases}$      ▷ early *explore*, late *exploit*
4:     Sample $(C_t, c_t^*)$ from $D$
5:     Current set $\mathcal{A}_{\mathrm{cur}} \leftarrow \begin{cases} \mathcal{A}_0, & t = 1 \\ \mathbf{Top}_k(\mathcal{M}) & \text{(by historical scores)} \end{cases}$
6:     **for all** $A_j \in \mathcal{A}_{\mathrm{cur}}$ **do**
7:         $R_j \leftarrow \text{SCORE}(A_j; C_t, c_t^*)$                  ▷ perform retrieval and similarity scoring
8:     **end for**
9:     $\mathcal{M} \leftarrow \mathcal{M} \cup \{(A_j, R_j)\}$
10:    $\mathcal{A}_{\mathrm{cur}} \leftarrow \mathbf{Top}_r(\mathcal{A}_{\mathrm{cur}}; R_j)$
11:    **for** $s = 1$ to $S$ **do**                                         ▷ Inner evolution
12:       $\mathcal{A}_{\mathrm{new}} \leftarrow \emptyset$
13:       **for all** $A_j \in \mathcal{A}_{\mathrm{cur}}$ **do**
14:         **for** $m = 1$ to $n$ **do**
15:           $A_{j,m} \leftarrow \begin{cases} \text{LOCALMUTATION}(A_j), & \text{Bernoulli}(p_t) = 1 \\ \text{GLOBALEXPLORATION}(A_j), & \text{otherwise} \end{cases}$
16:           $\mathcal{A}_{\mathrm{new}} \leftarrow \mathcal{A}_{\mathrm{new}} \cup \{A_{j,m}\}$
17:         **end for**
18:       **end for**
19:       **for all** $A \in \mathcal{A}_{\mathrm{new}}$ **do**
20:         $R \leftarrow \text{SCORE}(A; C_t, c_t^*)$
21:       **end for**
22:       $\mathcal{M} \leftarrow \mathcal{M} \cup \{(A, R) : A \in \mathcal{A}_{\mathrm{new}}\}$
23:       $\mathcal{A}_{\mathrm{cur}} \leftarrow \mathbf{Top}_r(\mathcal{A}_{\mathrm{new}}; R)$
24:    **end for**
25: **end for**
26: $A^* \leftarrow \arg\max_{A \in \mathcal{M}} R$
27: **return** $A^*$

---

generates $k$ agents as the initial search set. In each outer iteration, the algorithm draws a sample pair $(C_t, c_t^*)$ from the training dataset. If it is the first iteration, the framework directly uses the initialized agents. For example, an initial agent may be assigned the retrieval prompt *"You are an information processing expert who can locate relevant content in long texts based on problems"*, with parameter settings including `temperature = 0.7`, `top_p = 0.8`, `frequency_penalty = 0.0`, `presence_penalty = 0.0`, and `max_tokens = 1024`. Otherwise, the algorithm forms the current set $\mathcal{A}_{\mathrm{cur}}$ by selecting the top-$k$ agents from the memory pool according to their historical scores. Each agent in $\mathcal{A}_{\mathrm{cur}}$ performs retrieval and the algorithm evaluates it against the reference context $c_t^*$ using a similarity function. The framework stores the resulting scores in the memory pool and retains the top-$r$ agents as the starting point for the inner evolution process. During inner evolution, each agent generates $n$ new candidates: with probability $p_t$, the algorithm applies local search (e.g., minor semantic modifications of the prompt and small perturbations to parameters); with probability $1 - p_t$, the algorithm applies global exploration (prompt restructuring using an LLM and large-scale parameter sampling). The framework evaluates these new agents, adds them to the memory pool, and preserves the top-$r$ agents for the next iteration. After completing all outer iterations, the algorithm selects the highest-scoring agent from the memory pool as the final output $A^*$.

## A.3 NOTATION

Table 6: Table of Mathematical Symbols

| Symbol | Description |
|---|---|
| **Dataset and Samples** | |
| $D$ | Training dataset of sample pairs $(C_i, c_i^*)$, $i = 1, \ldots, N$. |
| $(C_i, c_i^*)$ | Sample pair with input $C_i$ and gold-standard context $c_i^*$. |
| $c_i^*$ | Gold-standard context for sample $C_i$. |
| **Algorithm Parameters** | |
| $k$ | Number of initialized agents. |
| $T_{\max}$ | Maximum number of outer iterations. |
| $S$ | Number of inner iterations for optimization. |
| $n$ | Number of searches (mutations) per agent. |
| $r$ | Number of top agents retained per round. |
| $p_t$ | Probability of local search at outer iteration $t$, dynamically scheduled between $p_{\min}$ and $p_{\max}$. |
| $p_{\min}, p_{\max}$ | Minimum and maximum local search probabilities used in scheduling. |
| **Agents and Memory** | |
| $\mathcal{M}$ | Memory pool storing agents and their scores, $\mathcal{M} = \{(A_j, s_{i,j})\}_{j=1}^M$. |
| $\mathcal{A}_0$ | Initial set of $k$ randomly generated agents, $\mathcal{A}_0 = \{A_j\}_{j=1}^k$. |
| $\mathcal{A}_{\text{cur}}$ | Current set of active agents. |
| $\mathcal{A}_{\text{new}}$ | Newly generated agents during inner evolution. |
| $A_j$ | An agent represented as $A_j = (C_j, \theta_j)$, with prompt $C_j$ and parameter vector $\theta_j$. |
| $C_j$ | prompt used by agent $A_j$. |
| $\theta_j$ | Parameter vector of agent $A_j$, $\theta_j \in \mathbb{R}^d$. |
| $\theta_j \sim \mathcal{N}(\mu_0, \sigma_0^2)$ | Initialization of $\theta_j$ from a normal distribution with mean $\mu_0$ and variance $\sigma_0^2$. |
| $A^*$ | Best-performing agent selected from $\mathcal{M}$. |
| **Scoring and Selection** | |
| $R_j$ | Score of agent $A_j$ on sample $C_t$, compared to $c_t^*$. |
| $s_{i,j}$ | Cosine similarity between retrieved context $c_{i,j}$ and gold-standard context $c_i^*$. |
| $c_{i,j}$ | Candidate context retrieved by agent $A_j$ for input $C_i$. |
| $\text{Sim}(c_{i,j}, c_i^*)$ | Cosine similarity, $\text{Sim}(c_{i,j}, c_i^*) = \frac{c_{i,j} \cdot c_i^*}{\|c_{i,j}\| \|c_i^*\|}$. |
| $\mathcal{A}_{\text{top-}k}$ | Top-$k$ agents by score, $\mathcal{A}_{\text{top-}k} = \arg\max_{A_j \in \mathcal{A}, |\mathcal{A}_{\text{top-}k}| = k} s_{i,j}$. |
| **Local and Global Search** | |
| $C_j''$ | Modified prompt after search. |
| $\epsilon_c^{(l)}$ | Semantic modification applied in local search. |
| $\epsilon_\theta^{(l)}$ | Local parameter perturbation, $\epsilon_\theta^{(l)} \sim \text{TruncNorm}(0, \sigma_l^2, [-k_l \sigma_l, k_l \sigma_l])$. |
| $\sigma_l^2$ | Variance of local search perturbations. |
| $k_l$ | Range constraint for local perturbations. |
| $\epsilon_c^{(g)}$ | Semantic modification applied in global exploration. |
| $\epsilon_\theta^{(g)}$ | Global parameter perturbation, $\epsilon_\theta^{(g)} \sim \text{TruncNorm}(0, \sigma_g^2, [-k_g \sigma_g, -\delta] \cup [\delta, k_g \sigma_g])$. |
| $\sigma_g^2$ | Variance of global search perturbations. |
| $k_g$ | Range constraint for global perturbations. |
| $\delta$ | Minimum perturbation magnitude for global search. |

## A.4 DETAILED EXPERIMENTAL SETUP

**Datasets.** We evaluate our methods on two groups of benchmarks. (1) *QA benchmarks:* Long-Bench Bai et al. (2023) targets long-context understanding with inputs up to 100K tokens. HotpotQA Yang et al. (2018) is a multi-hop QA dataset over multiple supporting documents. 2WikiMultihopQA Ho et al. (2020) focuses on compositional multi-hop questions in Wikipedia. MuSiQue Trivedi et al. (2022) decomposes multi-hop reasoning into single-hop questions. Natural Questions (NQ) Kwiatkowski et al. (2019) contains real queries with Wikipedia passages. SQuAD Rajpurkar et al. (2016) is a reading comprehension dataset with span-based answers.

(2) *Reasoning benchmarks:* MATH500 Hendrycks et al. (2021) is a subset of the MATH dataset containing 500 challenging competition-level problems. MMLU-pro Wang et al. (2024) extends the standard MMLU benchmark with more professional and domain-specific tasks. GPQA Rein et al. (2024) evaluates graduate-level problem-solving ability in science and engineering domains. DROP Dua et al. (2019) is a reading comprehension dataset that requires discrete reasoning such as addition, counting, and comparison. MuSR Sprague et al. (2023) is a benchmark for multi-step symbolic reasoning, which contains two subsets: MuSR-op, focusing on operational reasoning, and MuSR-ta, targeting task abstraction.

**LLM Models.** We experiment with a range of proprietary and open-source LLMs: GPT-4o-mini and GPT-4.1-nano, representing compact yet strong proprietary models; Qwen-Long and Qwen-Max (Qwen team), designed for long-context reasoning and high performance; and Gemma2-27B (Google DeepMind), a large open-source model optimized for efficiency and accuracy.

**Baselines.** We compare RAG-FGO with two groups of representative methods. (1) *Retrieval-based baselines:* **Direct** lets the LLM answer questions without retrieval. **ReAct** Yao et al. (2023) integrates chain-of-thought reasoning with retrieval tool use. **Self-Act** Press et al. (2022) (Self-Ask with Search) decomposes a query into sub-questions and retrieves evidence for each. **Aflow** Zhang et al. (2024a) employs an adaptive flow strategy that dynamically decides when to reason or retrieve, making it effective for multi-hop QA. (2) *Reasoning-based baselines:* **Zero-shot CoT** elicits reasoning by adding a trigger phrase (e.g., "Let us reason step by step") to the query without in-context exemplars. **Few-shot CoT** guides reasoning by providing a small set of annotated exemplars, allowing the model to learn from in-context demonstrations. **USC** Chen et al. (2023) generates multiple CoT reasoning paths and selects the final answer by majority voting. **CoT-WP** Wang & Zhou (2024) weights reasoning paths by confidence scores from token-level probabilities and aggregates them via weighted voting. **LEAP** Zhang et al. (2024b) enhances few-shot prompting by inducing errors on exemplars and using self-reflection to extract transferable reasoning principles. **ReFeri** Lee et al. (2025a) recycles few-shot exemplars to score and verify candidate outputs, selecting responses that are both confident and contextually coherent without additional training.

## A.5 DEMONSTRATION OF RESULTS

### A.5.1 BASELINE CONTEXTUAL RETRIEVAL PERFORMANCE

---

**Contextual Information and Selection(Without Optimized prompt)**

**Problem:**

Which organization declared the First Year of Studies program at Notre Dame "outstanding?

**Context:**

Architecturally, the school has a Catholic character. Atop the Main Building's gold dome is a golden statue of the Virgin Mary. Immediately in front of the Main Building and facing it, is a copper statue of Christ with arms upraised with the legend "Venite Ad Me Omnes". . .
. . .
**The program also includes a Learning Resource Center which provides time management, collaborative learning, and subject tutoring. This program has been recognized previously, by U.S. News & World Report, as outstanding.** (✓) In 2015-2016, Notre Dame ranked 18th overall among "national universities" in the United States in U.S. News & World Report's Best Colleges 2016. U.S. News & World Report also lists Notre Dame

---

Law School as 22nd overall...

...

All of Notre Dame's undergraduate students are a part of one of the five undergraduate colleges at the school or are in the First Year of Studies program.**The First Year of Studies program was established in 1962 to guide incoming freshmen in their first year at the school before they have declared a major.(✗)** Each student is given an academic advisor...

...

**Answer:**

**The university is the major seat of the Congregation of Holy Cross(✗)**

### A.5.2 FORWARD INFERENCE EXAMPLES

In this section, we initialize the first iteration of the retrieval-evaluation process. The experiment centers around the following question:

> "Which organization declared the First Year of Studies program at Notre Dame 'outstanding?'"

We employ HuggingFaceEmbeddings with the sentence-transformers/all-mpnet-base-v2 model as our embedding encoder. This model is widely adopted for semantic retrieval tasks due to its strong performance in capturing sentence-level meaning.

In addition, we load a precomputed vector file, vector_squad_results.npy, which contains evaluation embeddings of the reference documents. These vectors are used to compute semantic similarity with the query embedding, providing the foundation for the subsequent retrieval stage.

**Initializing iteration 1**

**Problem:** YYYY-MM-DD 11:56:30,047 - log.logger_utils - INFO
Which organization declared the First Year of Studies program at Notre Dame "outstanding?

**HuggingFaceEmbeddings with model:** YYYY-MM-DD 11:56:30,047 - log.logger_utils - INFO
sentence-transformers/all-mpnet-base-v2

**evaluation vectors:** YYYY-MM-DD 11:56:45,762-log.logger_utils-INFO
load docs/eval/vector_squad_results.npy

In the first iteration loop, the system dynamically generates processing parameters and heuristic prompts to guide the retrieval process. Randomized decoding parameters are sampled, including: temperature (0.60), top_p (0.87), frequency penalty (1.10), presence penalty (0.99), and max tokens (1219). These settings help simulate variability in generative behavior.

A contextual prompt is also generated to enhance retrieval alignment:

> "Look for key terms, dates, and names that are emphasized or repeated in the text. Focus on sections that discuss specific processes, outcomes, or examples. Pay attention to any questions posed within the text as they often highlight important concepts."

The prompt is vectorized using the same sentence-transformer model and then compared against precomputed evaluation vectors. A similarity score of 4.46 is computed, serving as a quantitative metric for retrieval relevance in this iteration.

---

**Initial loop 1 of iteration 1**

**First iteration. Generating dynamic parameters and prompt:** YYYY-MM-DD
11:56:45,763 - log.logger_utils - INFO

**Generating random settings for message processing:** YYYY-MM-DD 11:56:45,763 -
log.logger_utils - INFO

**Generated random settings:** YYYY-MM-DD 11:56:45,764-log.logger_utils - INFO
temperature: 0.601128214808998
top_p: 0.8672504958566317
frequency_penalty: 1.100216859549417
presence_penalty: 0.9924432136822198
max_tokens: 1219

**Generated prompt for first iteration:** YYYY-MM-DD
11:56:55,910-log.logger_utils-INFO
"Look for key terms, dates, and names that are emphasized or repeated in the text. Focus
on sections that discuss specific processes, outcomes, or examples. Pay attention to any
questions posed within the text as they often highlight important concepts."

**Generated template for non-longbench datatype:** YYYY-MM-DD 11:56:55,911 -
log.logger_utils - INFO

**Prompt vectorized:** YYYY-MM-DD 11:56:56,945 - log.logger_utils - INFO

**Calculating similarity:** YYYY-MM-DD 11:56:56,946 - log.logger_utils - INFO
Calculating similarity between generated vector and evaluation vector
Calculated similarity score: 4.46

---

In the second loop of the first iteration, the system continues to generate processing parameters
and heuristic prompts to further optimize the retrieval process. The randomized decoding settings
include: temperature (0.50), top_p (0.80), frequency penalty (1.23), presence penalty (1.12), and
max tokens (1223). These parameters introduce more variability to the retrieval behavior.

Additionally, a new contextual prompt is generated to further refine the retrieval strategy:

> "Look for key phrases or terms that highlight specific details or concepts within
> the text. Focus on identifying these elements to guide your search and extract
> relevant information more effectively."

The prompt is vectorized and compared against the precomputed evaluation vectors. The calculated
similarity score of 4.51 serves as a quantitative metric for retrieval relevance.

---

**Initial loop 2 of iteration 1**

**First iteration. Generating dynamic parameters and prompt:** YYYY-MM-DD
11:56:56,955 - log.logger_utils - INFO

**Generating random settings for message processing:** YYYY-MM-DD 11:56:56,955 -
log.logger_utils - INF

**Generated random settings:** YYYY-MM-DD 11:56:56,955 - log.logger_utils - INFO
temperature: 0.4985684861761591
top_p: 0.8046081601782339
frequency_penalty: 1.2308277314623417
presence_penalty: 1.1221325932968838

---

max_tokens: 1223

**Generated prompt for first iteration:**   YYYY-MM-DD 11:57:05,415 - log.logger_utils - INFO

"Look for key phrases or terms that highlight specific details or concepts within the text. Focus on identifying these elements to guide your search and extract relevant information more effectively."

**Generated template for non-longbench datatype:**   YYYY-MM-DD 11:57:05,415 - log.logger_utils - INFO

**Prompt vectorized:**   YYYY-MM-DD 11:57:05,946 - log.logger_utils - INFO

**Calculating similarity:**   YYYY-MM-DD 11:57:05,947 - log.logger_utils - INFO
Calculating similarity between generated vector and evaluation vector
Calculated similarity score: 4.51

In the third loop of the first iteration, the system generates new processing parameters and a heuristic prompt to enhance the retrieval process. The randomized decoding settings include: temperature (0.62), top_p (0.85), frequency penalty (1.11), presence penalty (0.86), and max tokens (1384). These settings introduce more variability into the retrieval behavior, providing a diverse approach to the generation process.

In addition, a new contextual prompt is generated to guide the retrieval strategy:

> "Look for key phrases like 'central theme', 'main character development', and
> 'plot twist' to quickly pinpoint the most relevant sections of the text."

The prompt is vectorized using the same sentence-transformer model and compared with the pre-computed evaluation vectors. The calculated similarity score of 4.33 serves as a quantitative measure of retrieval relevance.

**Initial loop 3 of iteration 1**

**First iteration. Generating dynamic parameters and prompt:**   YYYY-MM-DD 11:57:05,963 - log.logger_utils - INFO

**Generating random settings for message processing:**   YYYY-MM-DD 11:57:05,963 - log.logger_utils - INFO

**Generated random settings:**   YYYY-MM-DD 11:57:05,964 - log.logger_utils - IN
temperature: 0.6242781696498141
top_p: 0.8502414182828493
frequency_penalty: 1.11080892331044
presence_penalty: 0.8611376170775371
max_tokens: 1384

**Generated prompt for first iteration:**   YYYY-MM-DD 11:57:14,761 - log.logger_utils - INFO

"Look for key phrases like 'central theme', 'main character development', and 'plot twist' to quickly pinpoint the most relevant sections of the text."

**Generated template for non-longbench datatype:**   YYYY-MM-DD 11:57:14,761 - log.logger_utils - INFO

**Prompt vectorized:**   YYYY-MM-DD 11:57:15,293 - log.logger_utils - INFO

> **Calculating similarity:** YYYY-MM-DD 11:57:15,293 - log.logger_utils - INFO
> Calculating similarity between generated vector and evaluation vector
> Calculated similarity score: 4.33

In the initial loop 4 of the first iteration, the system generates dynamic parameters and a contextual prompt to enhance the retrieval process. The randomized decoding settings include: temperature (0.37), top_p (0.78), frequency penalty (1.09), presence penalty (1.04), and max tokens (1604). These parameters introduce further variability to the retrieval process, influencing the overall message generation behavior.

In addition, a new contextual prompt is generated to refine the retrieval strategy:

> "Look for key phrases, dates, names, or specific events mentioned in the text that are directly related to the question at hand. Pay attention to any bolded or italicized words as they may highlight important information. Additionally, check for headings and subheadings that can guide you to relevant sections of the text."

The prompt is vectorized, and the similarity score between the generated vector and the precomputed evaluation vector is calculated. The calculated similarity score of 4.48 provides a quantitative measure of retrieval relevance

> **Initial loop 4 of iteration 1**
>
> **First iteration. Generating dynamic parameters and prompt.** YYYY-MM-DD 11:57:15,300 - log.logger_utils - INFO
>
> **Generating random settings for message processing.** YYYY-MM-DD 11:57:15,300 - log.logger_utils - INFO
>
> **Generated random settings:** YYYY-MM-DD 11:57:15,300 - log.logger_utils - INFO
> temperature: 0.3678920224612938
> top_p: 0.7764401217965291
> frequency_penalty: 1.0889837430738198
> presence_penalty: 1.0359176679908866
> max_tokens: 1604
>
> **Generated prompt for first iteration:** YYYY-MM-DD 11:57:24,855 - log.logger_utils - INFO
> "Look for key phrases, dates, names, or specific events mentioned in the text that are directly related to the question at hand. Pay attention to any bolded or italicized words as they may highlight important information. Additionally, check for headings and subheadings that can guide you to relevant sections of the text."
>
> **Generated template for non-longbench datatype** YYYY-MM-DD 11:57:24,856 - log.logger_utils - INFO
> **Prompt vectorized.** YYYY-MM-DD 11:57:25,406 - log.logger_utils - INFO
>
> **Calculating similarity.** YYYY-MM-DD 11:57:25,406 - log.logger_utils - INFO
> Calculating similarity between generated vector and evaluation vector.
> Calculated similarity score: 4.48

In the initial loop 4 of the first iteration, the system generates dynamic parameters and a contextual prompt to improve the retrieval process. The randomized decoding settings include: temperature (0.37), top_p (0.78), frequency penalty (1.09), presence penalty (1.04), and max tokens (1604). These settings introduce additional variability to the retrieval process, influencing the generated messages and their relevance.

Additionally, a new contextual prompt is created to further refine the retrieval strategy:

> "Look for key phrases, dates, names, or specific events mentioned in the text that are directly related to the question at hand. Pay attention to any bolded or italicized words as they may highlight important information. Additionally, check for headings and subheadings that can guide you to relevant sections of the text."

The prompt is then vectorized and compared with precomputed evaluation vectors. The calculated similarity score of 4.48 quantifies the relevance of the retrieval

---

**Initial loop 5 of iteration 1**

**First iteration. Generating dynamic parameters and prompt.**   YYYY-MM-DD 11:57:25,411 - log.logger_utils - INFO

**Generating random settings for message processing.**  YYYY-MM-DD 11:57:25,411 - log.logger_utils - INFO

**Generated random settings:**  YYYY-MM-DD 11:57:25,412 - log.logger_utils - INFO
temperature: 0.4135428226062874
top_p: 0.9001342756678229
frequency_penalty: 0.9620386820341362
presence_penalty: 1.104212780284055
max_tokens: 1551

**Generated prompt for first iteration:**  YYYY-MM-DD 11:57:33,204 - log.logger_utils - INFO
"Look for phrases like k̈ey informationänd k̈eywordsẗo find the relevant sections in the text."

**Generated template for non-longbench datatype**  YYYY-MM-DD 11:57:33,205 - log.logger_utils - INFO

**Prompt vectorized.**  YYYY-MM-DD 11:57:33,733 - log.logger_utils - INFO

**Calculating similarity:**  YYYY-MM-DD 11:57:33,733 - log.logger_utils - INFO
Calculating similarity between generated vector and evaluation vector.
Calculated similarity score: 4.23

---

In the initial initialization of all results, the system selects the top 3 results as the basis for subsequent optimization. The first-ranked result achieved the highest similarity score of 4.51. The second-ranked result received a similarity score of 4.48. The third-ranked result scored 4.46. These three results will serve as the foundation for further optimization.

---

**Top 3 results for iteration 1**

**Question:** Which organization declared the First Year of Studies program at Notre Dame "outstanding?"  YYYY-MM-DD 11:57:33,739 - log.logger_utils - INFO

**Rank First:**
prompt:
"Look for key phrases or terms that highlight specific details or concepts within the text. Focus on identifying these elements to guide your search and extract relevant information more effectively."
**Parameter:**
temperature: 0.4135428226062874
top_p: 0.9001342756678229

---

frequency_penalty: 0.9620386820341362
presence_penalty: 1.104212780284055
max_tokens: 1551

**Score:** 4.51

**Rank Second:**
**prompt:**
"Look for key phrases, dates, names, or specific events mentioned in the text that are directly related to the question at hand. Pay attention to any bolded or italicized words as they may highlight important information. Additionally, check for headings and subheadings that can guide you to relevant sections of the text."
**Parameter:**
temperature: 0.4135428226062874
top_p: 0.9001342756678229
frequency_penalty: 0.9620386820341362
presence_penalty: 1.104212780284055
max_tokens: 1551

**Score:** 4.48

**Rank Third:**
**prompt:**
"Look for key terms, dates, and names that are emphasized or repeated in the text. Focus on sections that discuss specific processes, outcomes, or examples. Pay attention to any questions posed within the text as they often highlight important concepts."
**Parameter:**
temperature: 0.4135428226062874
top_p: 0.9001342756678229
frequency_penalty: 0.9620386820341362
presence_penalty: 1.104212780284055
max_tokens: 1551

**Score:** 4.46

In the method, five independent variations were applied to each individual in the top 3, with each variation randomly selecting either the local search or global search method for adjustment. Below is the process for the variation of the top1 individual.

Initially, the prompt for top1 focused on key terms, dates, and names that were emphasized or repeated in the text, particularly in sections discussing specific processes, outcomes, or examples. The initial parameters were set as follows: temperature: 0.4135, top_p: 0.9001, frequenc_penalty: 0.9620, presence_penalty: 1.1042, max_tokens: 1551. In the first variation, the prompt was adjusted to focus on identifying important phrases, timelines, and figures, with special attention to any questions posed in the text. The parameters were updated to temperature: 0.5721, top_p: 0.5677, frequency_penalty: 1.4988, presence_penalty: 1.1443, max_tokens: 1520.

In the second variation, the prompt was further refined to concentrate on identifying important phrases, timelines, and figures, with an increased emphasis on questions within the text. The parameters were updated to temperature: 0.4772, top_p: 0.8961, frequency_penalty: 1.3055, presence_penalty: 0.9751, max_tokens: 1548. In the third variation, the prompt continued to focus on recurring elements and methods, with the parameters adjusted to temperature: 0.7317, top_p: 0.9898, frequency_penalty: 1.3101, presence_penalty: 1.0031, max_tokens: 1518.

In the fourth variation, the prompt was adjusted to focus on repeated terms and emphasized sections in the text, with parameters updated to temperature: 0.5636, top_p: 1.0, frequency_penalty: 1.2820, presence_penalty: 1.0436, max_tokens: 1659. In the final variation, the prompt was further optimized to focus on identifying highlighted or repeated terms in the text, with special attention

to the question sections. The parameters were adjusted to temperature: 0.5306, top_p: 0.9886, frequency_penalty: 1.3734, presence_penalty: 0.9978, max_tokens: 1185.

These independent variations demonstrate how the use of both local and global search methods allows for the gradual refinement of the prompt and parameters, thereby increasing the diversity of the search results and reducing the occurrence of local optima.

---

**Rank First — search method**

**1. Selected method of change: Global Search(huge_llm_rewrite)**     YYYY-MM-DD
11:57:44,936 - log.logger_utils - INFO

**prompt before changes:**     11:57:44,936
Look for key terms, dates, and names that are emphasized or repeated in the text. Focus on sections that discuss specific processes, outcomes, or examples. Pay attention to any questions posed within the text as they often highlight important concepts.
**Parameter before changes:**
temperature: 0.4135428226062874
top_p: 0.9001342756678229
frequency_penalty: 0.9620386820341362
presence_penalty: 1.104212780284055
max_tokens: 1551

**prompt after changes:**     11:57:49,466
Identify important phrases, timelines, and figures that stand out or recur throughout the passage. Concentrate on areas detailing particular methodologies, results, or case studies. Be mindful of inquiries embedded in the material since they frequently point to pivotal ideas.
**Parameter after changes:**
temperature: 0.5721068931861351,
top_p: 0.5676959400915083
frequency_penalty: 1.4987603874898492
presence_penalty: 1.1443258291917342
max_tokens: 1520

**2. Selected method of change: Global Search(huge_llm_rewrite)**     YYYY-MM-DD
11:57:52,538 - log.logger_utils - INFO

**prompt before changes:**     11:57:44,936
Look for key terms, dates, and names that are emphasized or repeated in the text. Focus on sections that discuss specific processes, outcomes, or examples. Pay attention to any questions posed within the text as they often highlight important concepts.
**Parameter before changes:**
temperature: 0.4135428226062874
top_p: 0.9001342756678229
frequency_penalty: 0.9620386820341362
presence_penalty: 1.104212780284055
max_tokens: 1551

**prompt after changes:**     11:57:49,466
Identify significant phrases, timelines, and figures that stand out or recur throughout the material. Concentrate on segments detailing particular methods, results, or instances. Note any inquiries within the content since they frequently underscore crucial ideas.
**Parameter after changes:**
temperature: 0.47719384205903226
top_p: 0.8960686776894129
frequency_penalty: 1.3054897877933571
presence_penalty: 0.9750842054843881

---

max_tokens: 1548

**3. Selected method of change: Local Search(synonym_replace)**    YYYY-MM-DD 11:57:52,538

**prompt before changes:**    11:57:44,936
Look for key terms, dates, and names that are emphasized or repeated in the text. Focus on sections that discuss specific processes, outcomes, or examples. Pay attention to any questions posed within the text as they often highlight important concepts.
**Parameter before changes:**
temperature: 0.4135428226062874
top_p: 0.9001342756678229
frequency_penalty: 0.9620386820341362
presence_penalty: 1.104212780284055
max_tokens: 1551

**prompt after changes:**    11:57:49,466
Identify significant phrases, timelines, and figures that stand out or recur throughout the material. Concentrate on segments detailing particular methods, results, or instances. Note any inquiries within the content since they frequently underscore crucial ideas.
**Parameter after changes:**
temperature: 0.7317032164992677
top_p: 0.9898444557181555
frequency_penalty: 1.3101336305424052
presence_penalty: 1.0031408744638104
max_tokens: 1518

**4. Selected method of change: Global Search(huge_llm_rewrite)**    YYYY-MM-DD 11:57:52,538

**prompt before changes:**    11:57:44,936
Look for key terms, dates, and names that are emphasized or repeated in the text. Focus on sections that discuss specific processes, outcomes, or examples. Pay attention to any questions posed within the text as they often highlight important concepts.
**Parameter before changes:**
temperature: 0.4135428226062874
top_p: 0.9001342756678229
frequency_penalty: 0.9620386820341362
presence_penalty: 1.104212780284055
max_tokens: 1551

**prompt after changes:**    11:57:56,017
Look for key terms, dates, and names that are stress or repeat in the text. Focalise on sections that discuss specific processes, outcomes, surgery examples. Pay aid to any questions posed inside the text as they oftentimes highlight important concepts.
**Parameter after changes:**
temperature: 0.563586878197027
top_p: 1.0
frequency_penalty: 1.2819543345022086
presence_penalty: 1.0435629114046694
max_tokens: 1659

**5. Selected method of change: Local Search(synonym_replace)**    11:57:52,538

**prompt before changes:**    11:57:44,936
Look for key terms, dates, and names that are emphasized or repeated in the text. Focus

on sections that discuss specific processes, outcomes, or examples. Pay attention to any questions posed within the text as they often highlight important concepts.
**Parameter before changes:**
temperature: 0.4135428226062874
top_p: 0.9001342756678229
frequency_penalty: 0.9620386820341362
presence_penalty: 1.104212780284055
max_tokens: 1551

**prompt after changes:**                                                          11:58:02,711
Look for key terms, dates, and names that are highlighted or repeated in the text. Focus on sections that discuss specific processes, outcomes, or examples. Pay attention to any questions posed within the text as they often highlight important concepts.
**Parameter after changes:**
temperature: 0.5305826874793012
top_p: 0.9886117174895845
frequency_penalty: 1.3734404809003928
presence_penalty: 0.9977642729206061
max_tokens: 1185

After applying five variations to each individual in the top 3, we performed a retrieval capability evaluation on all the resulting variants. Based on the results of the evaluation, the top 3 individuals were selected and passed into the next iteration for further refinement. This process was repeated in a cyclic manner, with the top 3 results being chosen after each evaluation to proceed to the subsequent iteration. At the end of the cycle, the prompt with the highest score was selected as the final result.

### A.5.3 EVALUATING EXAMPLES

The provided section demonstrates the retrieval results after applying the optimized prompt to the problem, "Which organization declared the First Year of Studies program at Notre Dame 'outstanding'?" The prompt used for retrieval focused on identifying crucial phrases, timelines, and individuals, while concentrating on areas detailing specific methodologies, results, or instances, and paying attention to embedded queries, which often point to pivotal ideas. The context provided included a description of Notre Dame's architecture and the Learning Resource Center, with particular emphasis on the program being recognized by "U.S. News & World Report" as outstanding. This key information directly addressed the query. The answer generated from this context, using the optimized prompt, was "U.S. News & World Report," effectively retrieving the relevant information and pinpointing the correct organization within the context. The blue-colored portion in the text represents the information retrieved using our method, which is consistent with the standard answer.

**Contextual Information and Selection(With Optimized prompt)**

**Problem:**
Which organization declared the First Year of Studies program at Notre Dame "outstanding"?

**Prompt(Top-Scoring Item in the Memory Pool):**
Identify crucial phrases, timelines, and individuals that stand out or recur. Concentrate on areas detailing particular methodologies, results, or instances. Notice queries embedded in the content, as they usually pinpoint pivotal ideas.

**Context:**
Architecturally, the school has a Catholic character. Atop the Main Building's gold dome is a golden statue of the Virgin Mary. Immediately in front of the Main Building and facing it, is a copper statue of Christ with arms upraised with the legend "Venite Ad Me Omnes..."
. . .

> **The program also includes a Learning Resource Center which provides time management, collaborative learning, and subject tutoring. This program has been recognized previously, by U.S. News & World Report, as outstanding.**(✓) In 2015-2016, Notre Dame ranked 18th overall among "national universities" in the United States in U.S. News & World Report's Best Colleges 2016. U.S. News & World Report also lists Notre Dame Law School as 22nd overall
> . . .
>
> **Answer:**
> **U.S. News & World Report**

### A.6    ACKNOWLEDGE

This article used large language models (such as ChatGPT) as an auxiliary tool in the language polishing process, but did not use them in research conception and academic content generation.

### A.7    ADDITIONAL METHODOLOGICAL DETAILS

#### A.7.1    OBJECTIVE FUNCTION AND CONVERGENCE INSIGHT

This section formally defines the optimization objective of RAG-FGO and provides a theoretical analysis of how the Fungal Growth Optimization (FGO) algorithm navigates the mixed-variable search space to ensure convergence through global exploration and elite preservation mechanisms.

**Formal Objective Definition.**    For any candidate agent $A = (C, \theta)$, the generated rewritten query $c(A, q)$ retrieves an evidence vector $h(c(A, q))$. We define the semantic similarity between this evidence and the reference supporting vector $h^{\text{ref}}$ as the scoring function:

$$R(A) = \text{Sim}\big(h(c(A, q)),\, h^{\text{ref}}\big),$$

where $\text{Sim}$ denotes cosine similarity. The optimization goal of RAG-FGO is therefore to identify the agent that maximizes this retrieval-quality score:

$$A^* = \arg\max_A R(A).$$

This formulation explicitly characterizes the criterion being optimized: the degree of semantic alignment between the retrieved evidence and the reference supporting content.

**Core Rationale.**    The objective of RAG-FGO is to maximize the semantic-similarity–based retrieval quality function $R(A) = \text{Sim}\big(h(c(A, q)), h^{\text{ref}}\big)$. The joint search space of prompt templates and decoding hyperparameters contains both *discrete* structural variables (e.g., prompt semantics) and *continuous* parameters (e.g., temperature, top-$p$). The objective is non-differentiable, non-convex, and noisy due to the stochastic nature of LLM generation. Prior work Audet et al. (2023); Talbi (2024) shows that such mixed-variable black-box spaces are difficult to optimize using gradient-based approaches or purely discrete local search. Motivated by these observations, RAG-FGO adopts a heuristic framework that integrates global exploration, local refinement, and elite preservation. These mechanisms complement each other and, in practice, increase the likelihood of discovering high-quality candidate strategies when optimizing $R(A)$.

**(1) Global Search (Spore Germination Mechanism).**    Global exploration applies structural prompt rewrites together with large-magnitude perturbations to a subset of candidates. This allows the search process to reach regions of the space that differ substantially in semantic style, generation mode, and parameter configuration. While this mechanism does not aim to exhaustively cover all potential high-value areas, it broadens the search coverage and empirically reduces the risk of becoming confined to a narrow region of the space.

**(2) Local Search (Hyphal Branching Mechanism).**    For candidates that perform well in the memory pool, RAG-FGO applies fine-grained semantic modifications (e.g., synonym-level rewrites, discourse refinements) and small-step parameter perturbations, generating a set of locally related but

complementary variants. All variants are re-evaluated, and those that achieve better scores or exhibit promising structural diversity are retained. Local search does not attempt to ensure that every mutation leads to an improvement; rather, it enables denser exploration around promising areas so that the search process can accumulate useful variants over multiple iterations. Empirically, this tends to increase the chance of uncovering better query reformulations without making strong assumptions about stepwise improvement.

**(3) Query Memory Pool (Elite Preservation Mechanism).** After each iteration, the memory pool retains a set of high-scoring candidates and uses them as references for subsequent search. The goal is not to enforce any strict monotonic improvement, but to prevent high-quality candidates from being lost due to stochastic perturbations and to provide a stable set of anchors for continued exploration. We also introduce diversity constraints on both prompt templates and parameter configurations to prevent the pool from collapsing into a single narrow region. In practice, this combination of elite preservation and diversity maintenance helps the search maintain momentum toward better-performing regions, though we do not claim any formal convergence guarantees.

**Empirical Validation** To validate the theoretical analysis, we compared RAG-FGO with several classic heuristic optimization algorithms (including PSO, GA, SA, etc.) under the same computational budget. As shown in Table 7, RAG-FGO achieved significant leads on the LongBench, HotpotQA, and 2Wiki datasets, outperforming the runner-up algorithm (FOA) by 2.7%, 5.1%, and 4.8%, respectively. While traditional evolutionary algorithms often converge slowly in mixed spaces and Simulated Annealing (SA) can get lost due to a lack of population memory, RAG-FGO successfully achieves efficient and stable convergence by combining global exploration with elite memory.

### A.7.2 PROMPT TEMPLATES FOR RAG-FGO SEARCH

```
Global Search Mutation Prompt Template

System Role:  You are a global-search rewrite agent.  Your task is to
perform a Global Exploration rewrite on the input retrieval prompt,
generating a structurally and semantically distinct Mutated Prompt.
You must follow these rewrite principles:

1.  Maximize Semantic Divergence:  Produce a rewritten prompt whose
semantic embedding is maximally distant from the original, enabling
the search to escape local optima.

2.  Enforce Structural Reconstruction:  Rewrite using entirely
different syntax, diction, persona, and narrative framing.  Minor
edits or synonym substitutions are strictly forbidden.

3.  Introduce a New Retrieval Paradigm:  Adopt a fundamentally
different retrieval or reasoning perspective (e.g., ``consistency
verification'', ``evidence triangulation'', ``multi-stage
decomposition'').

4.  Preserve Core Intent:  Maintain the original goal|improving
retrieval quality|while expressing it from a completely new
conceptual angle.

Input:
   Original Prompt:  {P}

Output:
   A rewritten Mutated Prompt that satisfies all principles above.
   Only output the rewritten prompt, with no explanations.
```

### A.7.3 CASE STUDY: EFFECTIVENESS OF GLOBAL SEARCH

To further illustrate the necessity of global search in RAG-FGO, we present a concrete example from HotpotQA where local perturbations fail to escape an incorrect semantic region, while global mutations successfully redirect retrieval.

**Example.** The original query:

> *"Which university did the author of The Golden Compass attend?"*

is frequently mapped to the film entity *The Golden Compass*, causing the retriever to return movie-related pages rather than information about the novel's author, Philip Pullman. Because local search only performs small-scale perturbations near the original embedding space, it remains trapped in this incorrect semantic region and consistently fails to locate the relevant biography pages. Global search, however, generates queries with substantially larger semantic shifts. Typical global mutations include:

- replacing the ambiguous title with the original book name *"Northern Lights"*;
- injecting explicit occupation cues such as *"author"* or *"writer"*;
- adding attribute-level signals such as *"educational background"*.

These large-scale rewrites shift the query vector from the film region into the literature–author–biographical region. This allows the retriever to retrieve the correct evidence, including the page stating that Philip Pullman attended Exeter College, Oxford. This case demonstrates that global search enables cross-region semantic jumps that local perturbations cannot achieve, and highlights its critical role in correcting retrieval failures caused by ambiguous or misleading query phrasing.

### A.8 EXTENDED EXPERIMENTS

#### A.8.1 EVALUATION AGAINST HEURISTIC OPTIMIZATION BASELINES

Table 7: Performance comparison of different heuristic optimizers on LongBench, HotpotQA, and 2WikiMultihopQA under identical RAG configurations (LLM: GPT-4o-mini).

| Method | LongBench | HotpotQA | 2WikiMultihopQA |
|---|---|---|---|
| PSO (Kennedy & Eberhart, 1995) | 66.2 | 72.9 | 69.1 |
| GA (Lambora et al., 2019) | 65.5 | 71.3 | 67.8 |
| SA  (Kirkpatrick et al., 1983) | 61.0 | 67.2 | 63.5 |
| FOA  (Ghaemi & Feizi-Derakhshi, 2014) | 68.8 | 76.4 | 72.5 |
| TLBO (Rao et al., 2011) | 63.3 | 69.0 | 65.4 |
| HS (Geem et al., 2001) | 64.1 | 70.2 | 66.7 |
| **RAG-FGO** | **71.5** | **81.5** | **77.3** |

We also compare RAG-FGO with several common heuristic optimizers, including PSO, GA, SA, FOA, TLBO, and HS, under the same RAG setup with GPT-4o-mini. Each method searches over the same prompt space and follows the same evaluation protocol. As shown in Table 7, FOA provides the strongest results among the heuristic baselines, while PSO and GA obtain moderate performance and SA, TLBO, and HS perform less competitively across datasets. RAG-FGO achieves the highest scores on all three benchmarks, indicating that its structured global–local search is more effective for RAG-oriented prompt optimization than general-purpose heuristic search.

#### A.8.2 EXPERIMENTAL COMPARISON WITH PROMPT OPTIMIZATION BASELINES

To situate RAG-FGO within the broader space of gradient-free and evolutionary prompt optimization techniques, we compare it against AutoPrompt, Genetic Prompt Search, and RLPrompt under identical RAG configurations. All methods use the same retriever and the Gemma-2-27B generator to ensure a controlled and comparable evaluation setting. Because these existing approaches

were originally developed for classification or masked-LM scenarios rather than retrieval-augmented generation with decoder-only models, we implement adapted, gradient-free variants suitable for query-rewriting prompts. As shown in Table 8, RAG-FGO achieves higher scores across HotpotQA, MuSiQue, and NQ. These results suggest that the performance differences arise from the structured global–local search procedure introduced in RAG-FGO rather than from generic prompt perturbation or search diversity alone.

Table 8: Comparison with gradient-free / evolutionary prompt optimization baselines. All results are reported in F1 (%), and all methods use the same retriever and Gemma-2-27B generator.

| Method | HotpotQA | MuSiQue | NQ | Avg. |
|---|---|---|---|---|
| AutoPrompt (Shin et al., 2020) | 60.2 | 54.5 | 61.3 | 58.6 |
| Genetic Prompt Search (Xu et al., 2022) | 58.7 | 63.9 | 67.9 | 63.5 |
| RLPrompt (Deng et al., 2022) | 65.1 | 59.2 | 65.4 | 63.2 |
| **RAG-FGO** | **74.5** | **66.1** | **73.1** | **71.2** |

### A.8.3 CROSS-TASK GENERALIZATION EXPERIMENTS

Table 9 reports the cross-task generalization results of RAG-FGO. In this setting, we optimize the retrieval agent on HotpotQA and directly apply it to single-hop QA datasets (NQ and SQuAD) without any additional tuning. All other RAG configurations and model settings remain identical to those used in the main experiments. The results show that the retrieval strategy optimized on

Table 9: Cross-task generalization of RAG-FGO from HotpotQA to single-hop QA datasets. All results are reported in F1.

| Target Dataset | Target-tuned | HotpotQA→Target | $\Delta$ |
|---|---|---|---|
| NQ | 73.8 | 71.1 | -2.7 |
| SQuAD | 79.2 | 77.3 | -1.9 |

the multi-hop HotpotQA dataset exhibits only limited performance degradation when transferred to single-hop QA tasks, with overall performance remaining relatively strong. This suggests that RAG-FGO does not rely on dataset-specific patterns, but instead learns more generalizable retrieval behaviors that transfer across task types.

### A.8.4 HYPERPARAMETER SENSITIVITY ANALYSIS

To assess the robustness of RAG-FGO under variations of key hyperparameters, we conduct a systematic sensitivity study on HotpotQA covering five categories of parameters: the threshold-related parameters $(\delta, \sigma_l)$, the local perturbation strength $k_l$, the global perturbation magnitude $\sigma_g$, and the retrieval scope $k_g$. Each parameter is varied independently within a reasonable range while all other settings are kept fixed, and results are reported as normalized changes relative to the default configuration.

Overall, RAG-FGO exhibits strong robustness across all five hyperparameter groups, with the default configuration achieving a stable balance between accuracy and computational cost. Specifically, the threshold parameters $(\delta, \sigma_l)$ show that excessively small values lead to overly conservative filtering and reduced usable evidence, whereas overly large values may introduce noise and increase overhead; the default setting consistently lies within a stable middle region. For the local perturbation strength $k_l$, small values restrict the exploration of local neighborhoods, while overly large values may generate unstable or low-quality perturbations; the default value provides a reasonable trade-off between exploration sufficiency and perturbation quality. Regarding the global perturbation magnitude $\sigma_g$, larger values can occasionally yield slight performance gains but also incur noticeably higher token costs, making the default setting more advantageous from an efficiency–effectiveness perspective. The retrieval scope parameter $k_g$ tends to degrade answer accuracy when set too high due to the introduction of irrelevant information, whereas moderately small values demonstrate more

Table 10: Sensitivity of hyperparameters $\delta, \sigma_l, \sigma_g, k_l, k_g$ on normalized *Avg F1* and *Cost*. All results are reported relative to the default configuration ($\delta = 1.00$, $\sigma_l = 1.00$, $\sigma_g = 0.50$, $k_l = 0.50$, $k_g = 0.10$).

| Param | Value | Avg F1 | Cost |
|---|---|---|---|
| $\delta$ | 0.50 | ↓0.8–1.5% | ↓2–4% |
| | 0.75 | ↓0.1–0.5% | ↓1–2% |
| | 1.00 | baseline | baseline |
| | 1.25 | ↑0.3–0.9% | ↑1–3% |
| | 1.50 | ↓0.5–1.2% | ↑1–3% |
| $\sigma_l$ | 0.50 | ↓0.6–1.2% | ↓2–4% |
| | 0.75 | ↓0.1–0.6% | ↓1–2% |
| | 1.00 | baseline | baseline |
| | 1.25 | ↑0.3–0.9% | ↑1–3% |
| | 1.50 | ↓0.5–1.3% | ↑1–3% |
| $\sigma_g$ | 0.00 | ↓1.0–1.5% | ↓4–7% |
| | 0.25 | ↓0.5–1.0% | ↓2–4% |
| | 0.50 | baseline | baseline |
| | 0.75 | ↑0.2–0.8% | ↑1–3% |
| | 1.00 | ↑0.8–1.5% | ↑3–5% |
| $k_l$ | 0.00 | ↓1.0–1.5% | ↓3–6% |
| | 0.25 | ↓0.4–0.9% | ↓2–4% |
| | 0.50 | baseline | baseline |
| | 0.75 | ↑0.3–0.9% | ↑1–3% |
| | 1.00 | ↓0.5–1.3% | ↑1–3% |
| $k_g$ | 0.05 | ↓0.1–0.6% | ↓1–3% |
| | 0.10 | baseline | baseline |
| | 0.20 | ↑0.3–1.0% | ↑2–5% |
| | 0.30 | ↓0.5–1.5% | ↑4–8% |

stable behavior and align well with the chosen default. Taken together, these results indicate that RAG-FGO exhibits limited sensitivity to hyperparameter choices within reasonable ranges, and can maintain strong, stable performance without extensive tuning. Table 10 summarizes the complete sensitivity results on normalized Avg F1 and Cost.

### A.8.5    INSTABILITY OF RAG-FGO WITHOUT THE QUERY MEMORY POOL

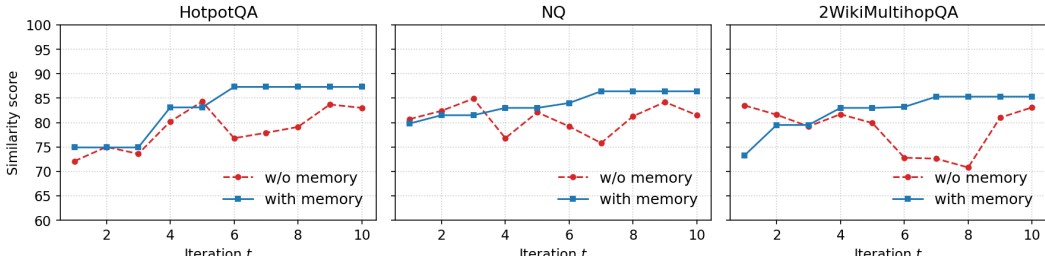

Figure 6: Semantic–similarity trajectories of RAG-FGO with and without the query memory pool.

To further examine the role of the query memory pool in shaping the search dynamics, we compare the evolution of semantic–similarity scores across outer iterations under the settings with and without the memory pool. As illustrated in Fig. 6, disabling the memory pool forces the algorithm to rely solely on the candidates generated in the current iteration. Because these candidates are affected by LLM stochasticity and perturbation magnitude, their scores exhibit substantial fluctuations, resulting in an unstable trajectory across iterations. In contrast, when the memory pool is enabled, each iteration evaluates both newly generated candidates and historically high-scoring ones, ensuring that the best-so-far solution is never lost due to random perturbations. While this mechanism does not impose any formal monotonicity or convergence guarantee, it consistently yields a smoother optimization trajectory in practice, with the best-so-far performance progressively moving toward higher-scoring regions. Overall, the query memory pool effectively suppresses stochastic volatility and enhances cross-iteration stability throughout the search process.

