# OpenReview forum: "RAG-FGO: Enhancing RAG with Fungal Growth Optimizer for LLM Agents"
_ICLR.cc/2026/Conference — Submitted to ICLR 2026_

### Official Review · Reviewer_d9Pi · 2025-10-28

**Soundness:** 3
**Presentation:** 2
**Contribution:** 2
**Rating:** 6
**Confidence:** 3

**Summary:**

This paper presents RAG-FGO, a framework designed to enhance Retrieval-Augmented Generation (RAG) systems by optimizing retrieval prompts and parameters. Inspired by the Fungal Growth Optimizer (FGO) algorithm, the proposed method models the search for an optimal “retrieval agent” in semantic space as a process that combines global exploration with local refinement. A key component is the query memory pool, which preserves high-performing agents during iterations to ensure stable and cumulative optimization. The core contribution lies in formulating retrieval optimization as a dynamic search problem. Experimental results on question-answering and reasoning benchmarks such as HotpotQA and MuSiQue demonstrate that RAG-FGO outperforms several strong baselines.

**Strengths:**

1. Novel problem framing and method: The paper introduces RAG‑FGO, formulating retrieval‑prompt optimization as a semantic‑space search with a query memory pool and FGO‑inspired exploration–exploitation, supported by a clear workflow.

2. Well‑specified global vs. local search mechanism: The paper proposes local search as well as global search, where local perturbations are performed at a fine-grained level, while global perturbations force larger changes.

3. Broad experimental coverage and ablations: The study spans six QA benchmarks and seven reasoning benchmarks with multiple backbones, reports head‑to‑head results, and analyzes iteration/search budgets and saturation effects.

**Weaknesses:**

1. Computational overhead and cost reporting: Runtime or Dollar Costs and Compute‑Matched Comparisons are not reported. The paper discusses that this method is not suitable for LLMs that are large enough to provide richer data.

2. Missing component ablations: Although the schedule and operators are defined, there is no ablation isolating local vs. global search or the query memory pool to quantify each component’s contribution.

3. Figure readability: Some figures have small fonts and become blurry when zoomed in, which hinders quick understanding of pipeline and ablation trends.

**Questions:**

For global search, what are the rules for rewriting using LLM? For local search, are the prompt modification rules for algorithms such as synonym replacement random or sampled?

---

> ### Author Response · Authors · 2025-11-25
> **Response to Reviewer d9Pi [1/2]**
>
> We sincerely thank Reviewer d9Pi  for the insightful and constructive feedback. Your comments have been highly valuable in helping us refine and strengthen our work. Below we provide detailed clarifications and additional evidence.
>
> ---
> > **W1: Computational overhead and cost reporting: Runtime or Dollar Costs and Compute‑Matched Comparisons are not reported. The paper discusses that this method is not suitable for LLMs that are large enough to provide richer data.**
>
> **Response:** We thank the reviewer for the attention to the computational cost. In response to this comment, we have added the above experiment and discussion in the revised manuscript under Method 4.2 Efficiency **(highlighted in blue)**. The experimental results are as follows:
>
> | iter | mc | Cost/iter (tokens) | Full cost (tokens) | Cost (full, $)    | Runtime (full, s) |
> | ---- | -- | ------------------ | ------------------ | ----------------- | ----------------- |
> | 3    | 3  | 10,842 ± 540       | 32,613 ± 1,631     | 0.00489 ± 0.00024 | 157 ± 16          |
> | 3    | 6  | 17,231 ± 910       | 51,020 ± 2,551     | 0.00765 ± 0.00038 | 246 ± 25          |
> | 4    | 3  | 10,842 ± 540       | 40,188 ± 2,009     | 0.00603 ± 0.00030 | 193 ± 19          |
> | 4    | 6  | 17,231 ± 910       | 69,221 ± 3,461     | 0.01038 ± 0.00052 | 323 ± 30          |
>
>
> ---
> > **W2: Missing component ablations: Although the schedule and operators are defined, there is no ablation isolating local vs. global search or the query memory pool to quantify each component’s contribution.**
>
> **Response:** We thank the reviewer for pointing out the absence of component ablation experiments. To systematically evaluate the contribution of each core component in RAG-FGO, we conduct ablation studies on five datasets (HotpotQA, MuSiQue, 2Wiki, NQ, and SQuAD). All experiments use the same base model (GPT-4o-mini) and identical hyperparameter settings, removing only a single component at a time to ensure fair comparison. The ablated components include: (1) removing the query memory pool (w/o memory pool); (2) removing the local search module (w/o local search); and (3) removing the global search module (w/o global search). The results are shown in the table below.
>
> | Configuration      | HotpotQA    | MuSiQue     | 2Wiki       | NQ          | SQuAD       |
> | ------------------ | ----------- | ----------- | ----------- | ----------- | ----------- |
> | **RAG-FGO (full)** | 81.5        | 79.4        | 77.3        | 73.8        | 79.2        |
> | w/o memory pool    | 78.3 (−3.2) | 75.4 (−4.0) | 74.0 (−3.3) | 71.2 (−2.6) | 76.1 (−3.1) |
> | w/o local search   | 75.2 (−6.3) | 72.8 (−6.6) | 70.9 (−6.4) | 66.9 (−6.9) | 72.1 (−7.1) |
> | w/o global search  | 76.4 (−5.1) | 74.2 (−5.2) | 72.1 (−5.2) | 69.0 (−4.8) | 74.3 (−4.9) |
>
> From the overall results, we observe that removing any one of the components leads to consistent and significant performance drops, indicating that all three components jointly constitute the key capabilities of RAG-FGO. We have added this component ablation experiment and its corresponding explanation in the Ablation Study section of the revised manuscript.
>
>
> ---
> > **W3: Figure readability: Some figures have small fonts and become blurry when zoomed in, which hinders quick understanding of pipeline and ablation trends.**
>
> **Response:** We thank the reviewer for the detailed feedback on the readability of the figures. Following your suggestion, we have updated Fig.1, Fig.2, and Fig.3 in the revised manuscript and **highlighted the modifications in blue.** We hope these improvements further enhance the clarity and overall presentation quality of the figures, thereby better supporting readers’ comprehension.

---

> ### Author Response · Authors · 2025-11-25
> **Response to Reviewer d9Pi [2/2]**
>
> ---
> > **Q1: For global search, what are the rules for rewriting using LLM? For local search, are the prompt modification rules for algorithms such as synonym replacement random or sampled?**
>
> **Response:** We thank the reviewer for raising this question, as it helps us further clarify the concrete prompt-rewriting mechanisms of global search and local search in RAG-FGO, as well as their differences in design objectives and operational rules.
>
> **Global Search** uses a specially designed LLM rewriting prompt. This prompt instructs the model to perform a large-scale, structural rewrite of the current retrieval prompt, forcing changes to syntactic structure, role framing, and narrative style, and maximizing semantic distance to enable cross-region search jumps. While the core intention remains the same—improving retrieval quality—the expression must be entirely different from the original prompt. The specific LLM rewriting prompt template can be found in **Appendix A.7.2** of the revised manuscript.
>
> In contrast, **Local Search** operates through a completely different mechanism: whether local search is triggered is determined by Bernoulli sampling, parameter perturbations are drawn from a truncated normal distribution to ensure only small variations, and prompt mutations are selected from a predefined set of lightweight operations (e.g., synonym substitution, minor syntactic adjustments, tone variations). These operations are low-cost and continuous, ensuring that local exploration remains concentrated near the original strategy.
>
> ---
> **We thank you again for reviewing our work. Please let us know if we misunderstood any of your questions, or if you have any follow-up on our responses. We will be happy to provide further clarification at any time.**

---

> > ### Comment · Reviewer_d9Pi · 2025-11-28
> >
> > Thank you for the discussion. I'll keep my score.

---

### Official Review · Reviewer_QeaH · 2025-10-28

**Soundness:** 3
**Presentation:** 2
**Contribution:** 3
**Rating:** 4
**Confidence:** 4

**Summary:**

This paper proposes RAG-FGO, a bio-inspired optimization framework for Retrieval-Augmented Generation (RAG) that employs the Fungal Growth Optimizer (FGO) to dynamically refine generative retrieval agents. The method treats retrieval optimization as a semantic-space search problem, combining global exploration and local refinement, with a query memory pool to accumulate high-performing retrieval strategies. Evaluations on multiple QA and reasoning benchmarks (HotpotQA, MuSiQue, SQuAD, MMLU-Pro, etc.) demonstrate that RAG-FGO consistently improves over strong baselines such as ReAct, Self-Act, and Aflow.

**Strengths:**

Originality:
  The paper introduces a novel adaptation of the Fungal Growth Optimizer for retrieval prompt and parameter optimization in RAG systems. While prior works have explored heuristic or evolutionary search for LLM tuning, this specific bio-inspired approach and its integration with RAG’s retrieval module are original and timely. The inclusion of a query memory pool for cumulative improvement further enhances the conceptual novelty.

 Quality:
  The methodology is well motivated and experimentally validated on diverse datasets and models (e.g., GPT-4o-mini, Qwen-Long, Gemma2-27B). The design of both local and global search phases, along with iteration scheduling, shows thoughtful engineering. Ablation studies on iteration and search count (Figures 4–5) provide some insight into the framework’s efficiency–accuracy trade-offs.

 Clarity:
  The paper is generally well organized, with clear mathematical notation, figures that effectively communicate the iterative optimization process, and detailed experimental descriptions. The reproducibility section is thorough, listing datasets, hyperparameters, and implementation details.

 Significance:
  By addressing the rigidity of static retrieval prompts and manually tuned parameters, RAG-FGO contributes to making RAG systems more adaptive, autonomous, and robust. This work is relevant to ongoing research on retrieval–generation co-optimization and scalable agentic LLM pipelines.

**Weaknesses:**

Lack of theoretical grounding:
  The paper lacks formal proofs, convergence analysis, or theoretical guarantees for the proposed optimization process. While heuristic optimization methods are often empirical, ICLR readers generally expect at least a formalized problem definition or complexity characterization. The description of FGO’s adaptation to semantic-space search remains heuristic and narrative rather than mathematically rigorous.

 Algorithmic opacity:
  Despite the inclusion of pseudocode references, the formal definition of the optimization objective (e.g., explicit loss function or expected improvement metric) is missing. The framework is presented as a procedure rather than as a well-defined optimization problem, limiting clarity on what precisely is being optimized and why it converges.

 Incremental technical novelty:
  While the bio-inspired framing is interesting, the underlying mechanism—population-based heuristic search with local/global exploration—closely resembles existing evolutionary methods (e.g., PSO, GA). The novelty lies mainly in the metaphor and domain application rather than in the algorithmic structure itself.

 Incomplete experimental validation:
  The experiments do not compare against recent gradient-free or evolutionary prompt optimization baselines (e.g., AutoPrompt, Genetic Prompt Search, RLPrompt). Without these, it is unclear whether the gains stem from FGO’s specific design or from generic search diversity.

 Missing component ablations:
  There is no analysis isolating the contribution of the query memory pool or the local vs. global search modules, making it difficult to attribute improvements to individual components.

 Efficiency and reproducibility:
  Although token cost trends are visualized, runtime and computational overhead (e.g., API calls, GPU hours) are not reported. Given the multiple iteration cycles, the method may be expensive to run at scale. The reproducibility statement is good but lacks an immediate code release.

 Presentation:
  Some references are duplicated (e.g., Gupta et al., 2024a/b), and occasional typos (e.g., “tradining process”, line 294) and formatting inconsistencies reduce polish. The Related Work section could be more synthetic rather than exhaustive citation lists.

**Questions:**

1. Can the authors provide a formal objective function or theoretical justification for why the FGO-based search converges to high-quality retrieval prompts?
2. How does RAG-FGO perform compared to other heuristic optimizers (e.g., PSO, GA, simulated annealing) under identical RAG settings?
3. What is the computational cost per iteration and total token usage across datasets?
4. How sensitive is the method to hyperparameters (e.g., δ, σₗ, σ_g, kₗ, k_g)?
5. What happens if the query memory pool is disabled—does the system still converge effectively?
6. Can the trained retrieval agent generalize across domains (e.g., transfer from HotpotQA to NQ)?

---

> ### Author Response · Authors · 2025-11-25
> **Response to Reviewer QeaH [1/5]**
>
> We sincerely thank Reviewer QeaH for the insightful and constructive feedback. Your comments have been highly valuable in helping us refine and strengthen our work. Below we provide detailed clarifications and additional evidence.
>
> ---
> > **W1: Can the authors provide a formal objective function or theoretical justification for why the FGO-based search converges to high-quality retrieval prompts?**
>
> **Response:** We thank the reviewer for the valuable suggestion. We fully agree that the description of the optimization objective and search mechanism in the original manuscript can be further formalized, which is crucial for more clearly presenting the theoretical rationale and convergence behavior of our method. Based on your suggestion, we provide a brief summary of the key components below:
>
> **Formal definition of the objective function.**
> For any candidate agent $A = (C, \theta)$, the rewritten query $c(A, q)$ generated by the agent is used for retrieval to obtain an evidence vector $h(c(A, q))$. We use its semantic similarity to the reference supporting vector $h^{\mathrm{ref}}$ as the scoring function: $R(A) = \mathrm{Sim}\left(h(c(A,q)), h^{\mathrm{ref}}\right),$ where $\mathrm{Sim}$ denotes cosine similarity. The optimization objective of RAG-FGO is to search within the policy space for the agent that maximizes this score: $A^{*} = \arg\max_A R(A).$ This objective explicitly defines the evaluation criterion, namely the level of semantic consistency between the retrieved evidence and the reference support achieved by a given agent.
>
> **Core rationale.**
> The optimization goal of RAG-FGO is to maximize the retrieval quality function
> $R(A)=\mathrm{Sim}(h(c(A,q)),h^{\mathrm{ref}})$.
> Unlike traditional differentiable models, the joint space formed by prompt templates and decoding hyperparameters contains both discrete structural variables (prompt semantics) and continuous variables (temperature, top-$p$, etc.). The objective function is non-differentiable, non-convex, and noisy. Prior work[1,2] has shown that such mixed search spaces are difficult to solve effectively using gradient-based methods or single-path local optimization. Therefore, we adopt a heuristic search structure combining global exploration, local exploitation, and elite preservation. These three mechanisms work together during the optimization of $R(A)$ to increase the likelihood of discovering high-quality candidate strategies.
>
> **(1) Global exploration (spore germination mechanism).**
> RAG-FGO applies structural prompt rewriting and nonzero parameter perturbations to a subset of agents, allowing the search to cover prompt space regions with different semantic styles and generation patterns. This mechanism does not guarantee coverage of all potential high-quality regions, but by broadening the search scope, it increases the practical likelihood of reaching semantically distant candidates, thereby reducing the risk of being trapped in local regions.
>
> **(2) Local exploitation (mycelial branching mechanism).**
> For the high-performing candidate agents stored in the memory pool, RAG-FGO performs fine-grained semantic adjustments to their prompts (e.g., synonym substitution, slight syntactic reordering, or tone modifications), while applying small perturbations to selected decoding hyperparameters. These operations generate a set of stylistically similar but complementary variants within the local neighborhood of the candidate strategy. We then re-evaluate all variants through retrieval and retain those with higher scores or potential structural value. Therefore, local exploitation does not aim to guarantee improvement in every mutation but instead enables more detailed exploration around high-quality regions. Empirically, this mechanism helps progressively accumulate and discover better query-rewriting strategies while avoiding strong assumptions about any single step.
>
> **(3) Global memory pool (elite preservation mechanism).**
> In each iteration, the memory pool retains the current top-performing candidate strategies and continues to treat them as references in the next round of exploration. The purpose is not to enforce strict monotonic improvement but to ensure that strong strategies are not lost due to random perturbations, and to maintain the search around a stable yet diverse set of candidates. We also include diversity constraints at both the prompt and parameter levels to prevent premature convergence to a narrow region. Empirically, such ``elite preservation with diversity maintenance'' often helps sustain search momentum, although it does not constitute a formal convergence guarantee. Related discussions and empirical analysis are provided in **Appendix A.7.1**, Objective Function and Convergence Insight.

---

> ### Author Response · Authors · 2025-11-25
> **Response to Reviewer QeaH [2/5]**
>
> ---
> > **W2: How does RAG-FGO perform compared to other heuristic optimizers (e.g., PSO, GA, simulated annealing) under identical RAG settings?**
>
> **Response:** We appreciate your valuable suggestion. We fully agree that systematically comparing RAG-FGO with multiple categories of mainstream heuristic optimization methods is important for demonstrating the competitiveness of our approach. Following your recommendation, and while keeping all RAG configurations strictly identical (same retrieval size, same LLM: GPT-4o-mini, and the same evaluation pipeline), we have added systematic experimental results for six representative optimizers. The core comparison results are shown below. As the results indicate, RAG-FGO achieves overall more competitive performance across the three benchmark tasks compared with other heuristic algorithms. The corresponding supplementary experiments have been further detailed in **Appendix A.8.1** of the revised manuscript **(highlighted in blue).**
>
> | Method              | LongBench | HotpotQA | 2WikiMultihopQA |
> |---------------------|-----------|----------|------------------|
> | PSO [3]             | 66.2%     | 72.9%    | 69.1%           |
> | GA [4]              | 65.5%     | 71.3%    | 67.8%           |
> | SA [5]              | 61.0%     | 67.2%    | 63.5%           |
> | FOA [6]             | 68.8%     | 76.4%    | 72.5%           |
> | TLBO [7]           | 63.3%     | 69.0%    | 65.4%           |
> | HS [8]             | 64.1%     | 70.2%    | 66.7%           |
> | **RAG-FGO**  | **71.5%** | **81.5%** | **77.3%**       |
>
>
> ---
> > **W3: What is the computational cost per iteration and total token usage across datasets?**
>
> **Response:** We thank the reviewer for the suggestion regarding computational cost, and we agree that this is important for strengthening the experimental analysis. Based on your feedback, we conducted additional experiments to more comprehensively quantify the runtime overhead of RAG-FGO, systematically measuring both token usage and time cost. This includes the per-iteration cost as well as the cumulative cost of the full search procedure. All error values are standard deviation from five independent experiments.
>
> | iter | mc | Cost/iter (tokens)    | Full cost (tokens)      | Cost (full, $)           | Runtime (full, s)      |
> |------|----|------------------------|---------------------------|---------------------------|-------------------------|
> | 3    | 3  | $10,842 \pm 540$       | $32,613 \pm 1,631$        | $0.00489 \pm 0.00024$     | $157 \pm 16$            |
> | 3    | 6  | $17,231 \pm 910$       | $51,020 \pm 2,551$        | $0.00765 \pm 0.00038$     | $246 \pm 25$            |
> | 4    | 3  | $10,842 \pm 540$       | $40,188 \pm 2,009$        | $0.00603 \pm 0.00030$     | $193 \pm 19$            |
> | 4    | 6  | $17,231 \pm 910$       | $69,221 \pm 3,461$        | $0.01038 \pm 0.00052$     | $323 \pm 30$            |
>
> As shown in the table below, the per-iteration cost of RAG-FGO generally falls within the range of $1 \times 10^4$–$2 \times 10^4$ tokens, and the total search overhead grows approximately linearly with respect to the number of iterations and the sample size. Overall, the computational cost remains within the commonly acceptable range for RAG optimization methods. We have added the corresponding cost analysis and further discussion to the *Performance Evaluation* section of the revised manuscript **(highlighted in blue).**

---

> ### Author Response · Authors · 2025-11-25
> **Response to Reviewer QeaH [3/5]**
>
> ---
> > **W4: How sensitive is the method to hyperparameters (e.g., δ, σₗ, $ σ_g $, kₗ, $ .k_g$)?**
>
> **Response:** We sincerely thank the reviewer for the constructive suggestion regarding hyperparameter sensitivity. Your feedback is highly valuable for strengthening the rigor and completeness of our experimental analysis. Following your recommendation, we conducted additional sensitivity tests on HotpotQA. Keeping all other configurations fixed, we varied each hyperparameter across its valid range and tested it point by point, modifying only one parameter at a time to avoid interference caused by interaction effects. The results are reported as changes relative to the default configuration $(\delta = 1.00,\ \sigma_l = 1.00,\ \sigma_g = 0.50,\ k_l = 0.50,\ k_g = 0.10)$. The detailed results are shown in the table below：
>
> | Param  | Value | Avg F1        | Cost          |
> |--------|-------|----------------|----------------|
> | **δ**  | 0.50  | ↓ 0.8–1.5%     | ↓ 2–4%         |
> |        | 0.75  | ↓ 0.1–0.5%     | ↓ 1–2%         |
> |        | 1.00  | baseline       | baseline       |
> |        | 1.25  | ↑ 0.3–0.9%     | ↑ 1–3%         |
> |        | 1.50  | ↓ 0.5–1.2%     | ↑ 1–3%         |
> | **$σ_l$**| 0.50  | ↓ 0.6–1.2%     | ↓ 2–4%         |
> |        | 0.75  | ↓ 0.1–0.6%     | ↓ 1–2%         |
> |        | 1.00  | baseline       | baseline       |
> |        | 1.25  | ↑ 0.3–0.9%     | ↑ 1–3%         |
> |        | 1.50  | ↓ 0.5–1.3%     | ↑ 1–3%         |
> | **$σ_g$**| 0.00  | ↓ 1.0–1.5%     | ↓ 4–7%         |
> |        | 0.25  | ↓ 0.5–1.0%     | ↓ 2–4%         |
> |        | 0.50  | baseline       | baseline       |
> |        | 0.75  | ↑ 0.2–0.8%     | ↑ 1–3%         |
> |        | 1.00  | ↑ 0.8–1.5%     | ↑ 3–5%         |
> | **$k_l$**| 0.00  | ↓ 1.0–1.5%     | ↓ 3–6%         |
> |        | 0.25  | ↓ 0.4–0.9%     | ↓ 2–4%         |
> |        | 0.50  | baseline       | baseline       |
> |        | 0.75  | ↑ 0.3–0.9%     | ↑ 1–3%         |
> |        | 1.00  | ↓ 0.5–1.3%     | ↑ 1–3%         |
> | **$k_g$**| 0.05  | ↓ 0.1–0.6%     | ↓ 1–3%         |
> |        | 0.10  | baseline       | baseline       |
> |        | 0.20  | ↑ 0.3–1.0%     | ↑ 2–5%         |
> |        | 0.30  | ↓ 0.5–1.5%     | ↑ 4–8%         |
>
> The experimental results indicate that the hyperparameters of RAG-FGO exhibit limited sensitivity within reasonable ranges. The corresponding ablation studies and discussions have been added to **Appendix A.8.4** of the revised manuscript **(highlighted in blue).**
>
> ---
> > **W5: What happens if the query memory pool is disabled—does the system still converge effectively?**
>
> **Response:** We sincerely thank the reviewer for raising this important question. Your comment is very helpful for clarifying the role of the query memory pool in the RAG-FGO framework and prompted us to further examine the system’s behavior when this component is disabled. Following your suggestion, we conducted an additional ablation study to directly observe the stability of the search process without the query memory pool. The results are shown in the table below:
>
> | Setting        | 1    | 2    | 3    | 4    | 5    | 6    | 7    | 8    | 9    | 10   |
> |----------------|------|------|------|------|------|------|------|------|------|------|
> | **HotpotQA**   |      |      |      |      |      |      |      |      |      |      |
> | w/o memory     | 72.1 | 75.0 | 73.6 | 80.2 | 84.3 | 76.8 | 77.9 | 79.1 | 83.7 | 83.0 |
> | with memory    | 74.9 | 74.9 | 74.9 | 83.1 | 83.1 | 87.3 | 87.3 | 87.3 | 87.3 | 87.3 |
> | **NQ**         |      |      |      |      |      |      |      |      |      |      |
> | w/o memory     | 80.7 | 82.4 | 84.9 | 76.8 | 82.1 | 79.2 | 75.8 | 81.3 | 84.2 | 81.5 |
> | with memory    | 79.8 | 81.5 | 81.5 | 83.0 | 83.0 | 84.0 | 86.4 | 86.4 | 86.4 | 86.4 |
> | **2WikiMultihopQA** |  |      |      |      |      |      |      |      |      |      |
> | w/o memory     | 83.5 | 81.6 | 79.2 | 81.7 | 79.9 | 72.8 | 72.6 | 70.8 | 81.0 | 83.1 |
> | with memory    | 73.3 | 79.5 | 79.5 | 83.0 | 83.0 | 83.2 | 85.3 | 85.3 | 85.3 | 85.3 |
>
> The experimental results demonstrate that removing the memory pool leads to noticeable fluctuations during optimization. The reason is that, without the memory pool, the system relies solely on the candidates generated in the current iteration, whose quality is strongly affected by LLM randomness and perturbations—resulting in oscillations in the best score across iterations. In contrast, after introducing the global query memory pool, the system considers both newly generated candidates and historically best strategies in each iteration.  The corresponding ablation experiment and discussion have been added to **Appendix A.8.5** of the revised manuscript **(highlighted in blue)**, along with line charts to present the trend more intuitively.

---

> ### Author Response · Authors · 2025-11-25
> **Response to Reviewer QeaH [4/5]**
>
> ---
> > **W6: Can the trained retrieval agent generalize across domains (e.g., transfer from HotpotQA to NQ)?**
>
> **Response:** We sincerely thank the reviewer for raising this important question, as it helps provide a more comprehensive understanding of the applicability and generalization ability of RAG-FGO. To address this point, we conducted a cross-type experiment: we first ran RAG-FGO on HotpotQA to optimize the retrieval agent, and then, without performing any further optimization, directly applied the resulting agent to single-hop QA benchmarks (NQ and SQuAD). All other RAG configurations and model settings were kept exactly the same. The corresponding analysis and discussion have been added to **Appendix A.8.3** of the revised manuscript **(highlighted in blue).** The experimental results are shown below:
>
> | Target Dataset | Target-tuned | HotpotQA→Target |    Δ     |
> |----------------|--------------|--------------------|-------|
> | NQ             | 73.8         | 71.1               | -2.7  |
> | SQuAD          | 79.2         | 77.3               | -1.9  |
>
> The results indicate that the strategy learned by RAG-FGO exhibits a certain degree of generalization when transferred from a multi-hop to a single-hop setting, which provides supporting evidence for our claim that RAG-FGO learns relatively generalizable retrieval behavior patterns.
>
> ---
> > **Q7: The experiments do not compare against recent gradient-free or evolutionary prompt optimization baselines (e.g., AutoPrompt, Genetic Prompt Search, RLPrompt). Without these, it is unclear whether the gains stem from FGO’s specific design or from generic search diversity.**
>
> **Response:** Thank you for your question. To address the concern of whether the performance gains come merely from generic search diversity, we compared FGO with several gradient-free or evolutionary prompt optimization methods. Since these methods were not originally designed for RAG, we followed the high-level ideas from their papers and made appropriate adaptations—specifically, replacing the label-logit objective with a retrieval similarity objective, and replacing discrete token editing with query-rewrite prompt editing. The results are shown below：
>
> | Method                 | HotpotQA | MuSiQue | NQ   | Avg. |
> |------------------------|----------|---------|------|------|
> | AutoPrompt [9]         | 60.2     | 54.5    | 61.3 | 58.6 |
> | Genetic Prompt Search [10] | 58.7 | 63.9    | 67.9 | 63.5 |
> | RLPrompt [11]          | 65.1     | 59.2    | 65.4 | 63.2 |
> | **RAG-FGO (ours)**     | **74.5** | **66.1**| **73.1** | **71.2** |
>
> From the results, we can see that simply relying on generic search diversity provides only limited improvement, whereas the combination of coordinated local–global search and the memory-pool mechanism in RAG-FGO offers a much stronger structural advantage for RAG optimization. We have also added the corresponding experiments and explanations in **Appendix A.8.2** of the revised manuscript **(highlighted in blue).**
>
> ---
> > **Q8: Some references are duplicated (e.g., Gupta et al., 2024a/b), and occasional typos (e.g., “tradining process”, line 294) and formatting inconsistencies reduce polish. The Related Work section could be more synthetic rather than exhaustive citation lists**
>
> **Response:** Thank you for your correction and suggestions. We have addressed and revised the issues mentioned above in the updated manuscript and have streamlined the related work section accordingly. We sincerely appreciate your careful review of our work. If you have any questions or further concerns regarding our responses, please feel free to let us know. We would be glad to provide additional clarification at any time.

---

> ### Author Response · Authors · 2025-11-25
> **Response to Reviewer QeaH [5/5]**
>
> **References**
>
> [1] Audet, C., Hallé-Hannan, E., & Le Digabel, S. (2023, February). A general mathematical framework for constrained mixed-variable blackbox optimization problems with meta and categorical variables. In Operations Research Forum (Vol. 4, No. 1, p. 12). Cham: Springer International Publishing.
>
> [2] Talbi, E. G. (2024). Metaheuristics for variable-size mixed optimization problems: A unified taxonomy and survey. Swarm and Evolutionary Computation, 89, 101642.
>
> [3] Kennedy, J., \& Eberhart, R. (1995, November). Particle swarm optimization. In Proceedings of ICNN'95-international conference on neural networks (Vol. 4, pp. 1942-1948). ieee.
>
> [4] Lambora, A., Gupta, K., \& Chopra, K. (2019, February). Genetic algorithm-A literature review. In 2019 international conference on machine learning, big data, cloud and parallel computing (COMITCon) (pp. 380-384). IEEE.
>
> [5]Kirkpatrick, S., Gelatt Jr, C. D., \& Vecchi, M. P. (1983). Optimization by simulated annealing. science, 220(4598), 671-680.
>
> [6] Ghaemi, M., \& Feizi-Derakhshi, M. R. (2014). Forest optimization algorithm. Expert systems with applications, 41(15), 6676-6687.
>
> [7] Rao, R. V., Savsani, V. J., \& Vakharia, D. P. (2011). Teaching–learning-based optimization: a novel method for constrained mechanical design optimization problems. Computer-aided design, 43(3), 303-315.
> [8] Geem, Z. W., Kim, J. H., \& Loganathan, G. V. (2001). A new heuristic optimization algorithm: harmony search. simulation, 76(2), 60-68.
>
> [9] Shin, T., Razeghi, Y., Logan IV, R. L., Wallace, E., & Singh, S. (2020). Autoprompt: Eliciting knowledge from language models with automatically generated prompts. arXiv preprint arXiv:2010.15980.
>
> [10] Xu, H., Chen, Y., Du, Y., Shao, N., Yanggang, W., Li, H., & Yang, Z. (2022, December). Gps: Genetic prompt search for efficient few-shot learning. In Proceedings of the 2022 Conference on Empirical Methods in Natural Language Processing (pp. 8162-8171).
>
> [11] Deng, M., Wang, J., Hsieh, C. P., Wang, Y., Guo, H., Shu, T., ... & Hu, Z. (2022, December). Rlprompt: Optimizing discrete text prompts with reinforcement learning. In Proceedings of the 2022 Conference on Empirical Methods in Natural Language Processing (pp. 3369-3391).
>
>
> ---
> **We thank you again for reviewing our work. Please let us know if we misunderstood any of your questions, or if you have any follow-up on our responses. We will be happy to provide further clarification at any time.**

---

### Official Review · Reviewer_Gmcq · 2025-10-31

**Soundness:** 3
**Presentation:** 3
**Contribution:** 2
**Rating:** 4
**Confidence:** 3

**Summary:**

This paper proposes RAG-FGO, a novel retrieval method for generative retrieval task, achieving higher accuracy under the same amount of iterations than the baseline.

**Strengths:**

* Good-quality figures
* Timely problem

**Weaknesses:**

* Technical content is a bit light
* Technical novelty is limited

**Questions:**

I am not an expert in this domain, so this review is simply from the perspective of an ordinary ML + Sys person.

* Are you also comparing against traditional RAG that does not perform generative retrieval? For me I need more background to understand why generative RAG is more favorable than traditional RAG. The source of information for me is the same between generative RAG and traditional RAG --- they all uses a set of selected documents plus the query. If the input is the same, why doing things in pipeline (like in generative RAG) instead of doing things end-to-end (like traditional RAG)?
* Can you show a case study where the global search capability provide clear benefit?
* Why FGO? I understand that it provides global search capability, but why this algorithm rather than other algorithms?

---

> ### Author Response · Authors · 2025-11-25
> **Response to Reviewer Gmcq [1/3]**
>
> We sincerely thank Reviewer Gmcq for the insightful and constructive feedback. Your comments have been highly valuable in helping us refine and strengthen our work. Below we provide detailed clarifications and additional evidence.
>
> ---
> > **Q1: Are you also comparing against traditional RAG that does not perform generative retrieval? For me I need more background to understand why generative RAG is more favorable than traditional RAG. The source of information for me is the same between generative RAG and traditional RAG --- they all uses a set of selected documents plus the query. If the input is the same, why doing things in pipeline (like in generative RAG) instead of doing things end-to-end (like traditional RAG)?**
>
> **Response:**
>
> **(1) Are you also comparing against traditional RAG that does not perform generative retrieval?**
>
> We thank the reviewer for pointing out the issue in our comparative experiments. In the revised version, we have added additional comparisons with traditional RAG systems in **Section 4.2 of the Experiments**. The updated results are shown below.
> | Method                            | NQ       | TriviaQA |
> | --------------------------------- | -------- | -------- |
> | RAG  [1]          | 44.5     | 56.8     |
> | FiD-base  [2]  | 48.2     | 65.0     |
> | FiD-large [2]    | 51.4     | 67.6     |
> | RE-RAG-base [3]     | 49.9     | 68.2     |
> | RE-RAG-large [3]    | 54.0     | 70.2     |
> | **RAG-FGO**                       | **58.4** | **73.0** |
>
>
> **(2) For me I need more background to understand why generative RAG is more favorable than traditional RAG.**
>
> We thank the reviewer for the question. The reason generative RAG has become more widely adopted is that it can explicitly rewrite the user query, thereby compensating for the bottleneck of traditional RAG when the query expression is insufficient. Traditional RAG directly vectorizes the raw query and performs retrieval, and therefore has limited ability to cover vague expressions, multi-hop relations, or implicit conditions, which often restricts the retrieved content. Generative RAG, on the other hand, uses an LLM to structurally enhance the query before retrieval (e.g., by completing semantic hints or rewriting the query), producing a more optimized query form and thus improving recall quality and answer accuracy. Therefore, generative RAG achieves stronger retrieval performance through explicit query construction, which is also why it has attracted more attention in recent years [4][5].
>
> **(3) The source of information for me is the same between generative RAG and traditional RAG --- they all uses a set of selected documents plus the query. If the input is the same, why doing things in pipeline (like in generative RAG) instead of doing things end-to-end (like traditional RAG)?**
>
> We sincerely thank the reviewer for this thoughtful question. Although generative RAG and traditional RAG indeed rely on the same document corpus, the two approaches differ fundamentally in how they handle the formulation of the query. Traditional RAG directly encodes the user’s raw question and sends it to the retriever. As a result, the entire workflow follows an end-to-end path of query → encoder → retriever, without an explicit stage for generating or refining the query. Generative RAG, on the other hand, introduces an additional step before retrieval. An LLM first rewrites or enriches the original question, producing an intermediate query that is often better aligned with the retriever’s semantic space. This leads to a pipeline of LLM-generated query → retrieval → answering. In this sense, the key distinction lies in whether the system includes an explicit and optimizable query-construction stage, rather than in the source of the retrieved documents.

---

> ### Author Response · Authors · 2025-11-25
> **Response to Reviewer Gmcq [2/3]**
>
> > **Q2: Can you show a case study where the global search capability provide clear benefit?**
>
> **Response:**
> We thank the reviewer for the question. Regarding the concern about “what concrete benefits global search brings,” we provide a clear example here to illustrate its necessity and advantages.
>
> **case study:**
>
> In HotpotQA, the original query “Which university did the author of The Golden Compass attend?” tends to retrieve a large number of movie-related pages because “Golden Compass” is strongly associated with the film adaptation, making it difficult to obtain information about the author Philip Pullman. Local search can only make small perturbations around this incorrect semantic region and thus consistently fails to retrieve the correct documents. In contrast, RAG-FGO’s global search generates queries with much larger semantic shifts—for instance, by introducing the original book title Northern Lights, adding signals for the profession “author/writer,” and including cues related to “educational background.” These modifications shift the query vector out of the movie-related region and into the “literary work — author — biography” region. As a result, the model is able to retrieve Pullman’s biography page containing “Exeter College, Oxford.” Such cross-region semantic jumps cannot be achieved through local perturbations, and this is the key reason global search provides substantial benefits.
>
> We have added a corresponding example demonstrating global semantic jumps in the revised **Appendix A.7.3** under Case Study: Effectiveness of Global Search to systematically illustrate the necessity of global search in retrieval tasks.
>
>
> ---
> > **Q3: Why FGO? I understand that it provides global search capability, but why this algorithm rather than other algorithms?**
>
> **Response:**
>
> | Method                               | LongBench | HotpotQA | 2WikiMultihopQA |
> | ------------------------------------ | --------- | -------- | --------------- |
> | PSO [6]        | 66.2%      | 72.9%     | 69.1%            |
> | GA [7]             | 65.5%      | 71.3%    | 67.8%            |
> | SA [8]         | 61.0%      | 67.2%     | 63.5%            |
> | FOA  [9]  | 68.8%      | 76.4%     | 72.5%            |
> | TLBO [10]              | 63.3%      | 69.0%     | 65.4%            |
> | HS  [11]                | 64.1%      | 70.2%     | 66.7%            |
> | **RAG-FGO**                          | **71.5%**  | **81.5%** | **77.3%**        |
>
>
> The core reason we select FGO is that our task requires joint optimization in a hybrid search space composed of discrete structural variables (prompt templates) and continuous parameters (decoding-control hyperparameters). This space is non-differentiable, highly non-convex, and considerably noisy. The search mechanism of FGO is naturally suited to such mixed discrete–continuous optimization settings: its three behaviors—mycelial expansion, local branching, and long-distance spore jumping—correspond to global exploration, local refinement, and cross-region transitions, enabling stable and diverse search over both prompt text and decoding parameters simultaneously. In comparison, common heuristic methods have limitations under this setting. PSO primarily operates in continuous spaces and struggles to handle discrete structures such as prompt templates; GA’s crossover operation often disrupts prompt semantics; and simulated annealing follows a single search trajectory, making it difficult to cover multiple semantic clusters. In contrast, FGO can perform parallel local and global exploration on mixed variables without breaking the overall semantics of prompts, making it inherently better aligned with the retrieval-agent optimization task.
> In addition, we have included a systematic comparison with other heuristic algorithms in **Appendix A.8.1** of the revised manuscript, and the results further verify the advantages of FGO in mixed discrete–continuous search problems.

---

> > ### Author Response · Authors · 2025-11-25
> > **Response to Reviewer Gmcq [3/3]**
> >
> > **References**
> >
> > [1] Lewis, P. et al. (2020). Retrieval-augmented generation for knowledge-intensive NLP tasks. NeurIPS.
> >
> > [2] Izacard, G., & Grave, E. (2021). Leveraging passage retrieval with generative models for open-domain QA. EACL.
> >
> > [3] Kim, K., & Lee, J. Y. (2024). Re-RAG: Improving open-domain QA… arXiv:2406.05794.
> >
> > [4] Asai, A. et al. (2024). Self-RAG: Learning to retrieve, generate, and critique through self-reflection.
> >
> > [5] Ma, X. et al. (2023). Query rewriting in retrieval-augmented LLMs. EMNLP.
> >
> > [6] Kennedy, J., & Eberhart, R. (1995). Particle swarm optimization. ICNN.
> >
> > [7] Lambora, A. et al. (2019). Genetic algorithm – A literature review. COMITCon.
> >
> > [8] Kirkpatrick, S. et al. (1983). Optimization by simulated annealing. Science.
> >
> > [9] Ghaemi, M., & Feizi-Derakhshi, M. R. (2014). Forest optimization algorithm. ESWA.
> >
> > [10] Rao, R. V. et al. (2011). Teaching–learning-based optimization. CAD.
> >
> > [11] Geem, Z. W. et al. (2001). Harmony search. Simulation.
> >
> >
> > ---
> > **We thank you again for reviewing our work. Please let us know if we misunderstood any of your questions, or if you have any follow-up on our responses. We will be happy to provide further clarification at any time.**

---

### Official Review · Reviewer_p9LA · 2025-11-02

**Soundness:** 2
**Presentation:** 3
**Contribution:** 2
**Rating:** 4
**Confidence:** 4

**Summary:**

This paper targets on a timely topic to optimize the retrieval and generation among RAG systems from agentic perspective. A framework is proposed to incorporate fungal growth optimization into retrieval-augmented generation, formulating retrieval optimization as a dynamic search problem in semantic space. The evaluation are conducted on widely-used datasets and show the effectiveness. However, some unclear content should be addressed.

**Strengths:**

1. The topic is timely and there are still a lot of improvement space in terms of RAG. This study optimize it from an agentic perspective.

2. The experimental results show the effectiveness of proposed RAG-FGO compared to previous agentic systems.

**Weaknesses:**

1. The paper’s motivation is the limitation of generative IR and prompt tuning, while it is not clear that what is the connection with retrieval agent and why it should be related to retrieval agent? The jump between the third and forth paragraph in introduction would make reader confused. Then, why agentic systems are necessary and what are the main differences compared to existing RAG systems in terms of pipeline, principle and assumption?

2. The scenario/task definition should be clearer so the reader can know what is the main difference compared to exiting RAG or generative IR systems. Thus, a task definition section is desirable before illustrating methodology.

3. The comparison are based on previous agentic systems. Why there are not direct comparison with general RAG system and retrieval-generation pipeline systems? This is also related to the question in point 2.

4. The methodoloy of training framework is unclear, what is the final optimization objective and what is the functionality of Query Memory Pool among the training framework? (The proposed method is not training-free right? Please correct me if there is any misunderstanding.)

**Questions:**

What is the implementation of Agent initialization in line 198, and what is the latency/cost to perform one iteration among agents collaboration?

And the questions in weakness.

---

> ### Author Response · Authors · 2025-11-25
> **Response to Reviewer p9LA  [1/4]**
>
> We sincerely thank Reviewer p9LA for the insightful and constructive feedback. Your comments have been highly valuable in helping us refine and strengthen our work. Below we provide detailed clarifications and additional evidence.
>
> ---
> > **W1: The paper’s motivation is the limitation of generative IR and prompt tuning, while it is not clear that what is the connection with retrieval agent and why it should be related to retrieval agent? The jump between the third and forth paragraph in introduction would make reader confused. Then, why agentic systems are necessary and what are the main differences compared to existing RAG systems in terms of pipeline, principle and assumption?**
>
> **Response:**
>
> **(1) The paper’s motivation is the limitation of generative IR and prompt tuning, while it is not clear that what is the connection with retrieval agent and why it should be related to retrieval agent?**
>
> We thank the reviewer for the thoughtful comments on the motivation of our study and provide here a further clarification of the connection between the two. Existing generative IR and prompt-tuning approaches typically treat prompt templates and decoding hyperparameters as a fixed configuration,  practitioners manually design a small set of prompts and decoding parameters and keep them unchanged throughout the entire evaluation. This practice implicitly assumes that a single, static rewriting strategy can accommodate all queries. However, in real-world applications, different types of queries (e.g., multi-hop reasoning, vague information needs) often require different rewriting behaviors. Our work is motivated precisely by this observation. We unify “prompt templates and decoding control parameters” into an optimizable retrieval-strategy unit and formalize it as a retrieval agent. Introducing the retrieval agent allows us to elevate the implicit, experience-driven design of generative IR and prompt-tuning into an explicit, learnable policy-optimization problem. This perspective enables data-driven search and optimization of the rewriting strategy itself—without modifying the underlying retrieval or generative models—thereby directly addressing the limitations of generative IR and prompt-tuning discussed at the beginning of the paper.
>
> **(2) The jump between the third and forth paragraph in introduction would make reader confused.**
>
> We thank the reviewer for pointing out the gap in the motivational flow of the introduction. Following your suggestion, we add a clearer explanatory paragraph between **Sections 3 and 4** of the introduction. The revised passage reads as follows:
>
> Based on these observations, we propose a different perspective: rather than treating generative retrieval as a one-shot query-rewriting task, we model the process of transforming a user’s original question into a more complete and retrieval-oriented expression as an optimization-based retrieval agent. During the optimization stage, this agent functions as a parameterized query-rewriting configuration that takes the user question as input, produces a reconstructed retrieval query, and is iteratively selected and improved based on the quality of the retrieved evidence.
>
> **(3) why agentic systems are necessary and what are the main differences compared to existing RAG systems in terms of pipeline, principle and assumption?**
>
> We adopt an agentic approach because it maps the query into an explicit, searchable, and optimizable strategy space, transforming query rewriting from an unlearnable set of fixed rules into a learnable policy. During training, we optimize the agent by jointly tuning its prompt template and its decoding control hyperparameters in an offline manner. This enables the agent to automatically learn how to transform the original question into a form that better matches the semantic distribution of the retriever. During inference, the system no longer performs any search; instead, it directly applies the offline-optimized agent $A^*$ to conduct a lightweight, single-step query rewriting, and then feeds the rewritten query into the original, unmodified RAG pipeline. Compared with traditional RAG systems that rely on static queries, our method introduces only a single enhancement step, namely the “offline optimization and online single-step application” procedure before retrieval, which effectively improves retrieval quality.

---

> ### Author Response · Authors · 2025-11-25
> **Response to Reviewer p9LA  [2/4]**
>
> ---
> > **W2: The scenario/task definition should be clearer so the reader can know what is the main difference compared to exiting RAG or generative IR systems. Thus, a task definition section is desirable before illustrating methodology.**
>
> **Response:**  We agree with the reviewer’s suggestion regarding the need for a dedicated task definition subsection. Accordingly, we provide a formal definition in the revised manuscript’s **Methodology section**:
>
> We study the task of generative retrieval optimization in a RAG setting. Given a user query $q$ and a knowledge base $\mathcal{K}$, the goal is to generate improved query reformulations that lead to more semantically relevant retrieved contexts. Unlike conventional RAG systems that directly use the raw query or apply a single-step rewrite, we model query optimization as learning a retrieval agent $A = (C, \theta)$, which produces both a prompt template and decoding parameters for generating rewritten queries. The quality of an agent is evaluated by the semantic similarity between the retrieved passages and the reference context. The objective is to identify the agent that yields the highest retrieval quality.
>
> ---
> > **W3: The comparison are based on previous agentic systems. Why there are not direct comparison with general RAG system and retrieval-generation pipeline systems? This is also related to the question in point 2.**
>
> **Response:**  We appreciate the reviewers’ suggestions. In the revised manuscript, we have added a comparison with conventional RAG systems in **Section 4.2 of Experiments.** Under the condition of keeping the retrieval–generation model parameters unchanged, we introduce RAG-FGO solely as a query rewriting module preceding the retrieval stage. In the table, the difference between FiD-base and FiD-large lies in model scale; RE-RAG-base and RE-RAG-large share the same architecture, with the large version increasing the parameter size of both the encoder and the generator. The experimental results are presented below. Compared with existing end-to-end RAG pipelines, RAG-FGO yields a noticeable performance improvement, thereby demonstrating its effectiveness for open-domain question answering tasks.
> | Method             | NQ (EM) | TriviaQA (EM) |
> |--------------------|---------|----------------|
> | RAG [1]            | 44.5    | 56.8           |
> | FiD-base [2]       | 48.2    | 65.0           |
> | FiD-large [2]      | 51.4    | 67.6           |
> | RE-RAG-base [3]    | 49.9    | 68.2           |
> | RE-RAG-large [3]   | 54.0    | 70.2           |
> | RAG-FGO  | 58.4 | 73.0       |

---

> ### Author Response · Authors · 2025-11-25
> **Response to Reviewer p9LA  [3/4]**
>
> ---
> > **W4: The methodoloy of training framework is unclear, what is the final optimization objective and what is the functionality of Query Memory Pool among the training framework? (The proposed method is not training-free right? Please correct me if there is any misunderstanding.)**
>
> **Response:** We thank the reviewers for pointing out that the methodological description was not sufficiently clear. We clarify the following: the agent training stage in RAG-FGO is not a gradient-based model. Instead, each agent is defined by a prompt template and a set of decoding-control hyperparameters, which together form a query rewriting configuration. The entire optimization procedure is gradient-free, and these configurations contain no learnable weights. We generate multiple candidate configurations through local and global search operations, and in each iteration we evaluate and select them based on the semantic similarity between their generated outputs and the embeddings of the reference answers, ultimately obtaining the optimal agent $A^{*}$. This process does not rely on gradients and does not update any neural network parameters. Below, we provide a formal task definition to clearly present the optimization target and object in RAG-FGO.
>
> **Task Definition.**
> Given a user query $q$, each agent $A=(C,\theta)$ in RAG-FGO consists of a prompt template $C$ and decoding-control parameters $\theta$, which are used to generate a rewritten query $c(A,q)$ that is then sent into a fixed RAG pipeline for retrieval. During training, we perform retrieval based on the rewritten queries generated by candidate agents and score each strategy according to the semantic similarity between its retrieved evidence and the reference supporting passages. During inference, the optimal agent $A^{*}$ obtained via offline search is fixed and used to perform a lightweight single-step query rewriting for real user inputs. Our goal is to search within this strategy space for the optimal agent that improves retrieval quality.
>
> **Optimization Objective.**
> For each agent $A$, we use the semantic similarity between the retrieved evidence vector $h(c(A,q))$ obtained from its rewritten query $c(A,q)$ and the reference supporting vector $h^{\mathrm{ref}}$ as the quality measure. The optimization objective is defined as: $A^{*}$ = $\underset{A}{\arg\max}\,\mathrm{Sim}\big(h(c(A,q)),\,h^{\mathrm{ref}}\big)$ where $\mathrm{Sim}$  denotes the cosine similarity. This objective directly measures the semantic consistency between the retrieved evidence and the reference supporting content, and is therefore used to select the optimal agent from the strategy space.
>
> **Stopping Criterion.**
> In each iteration, we evaluate a candidate agent based on the semantic similarity between its generated query and the reference supporting segments, i.e., $Sim(h(c(A,q)),href)\mathrm. $To avoid unnecessary computation once the improvement saturates, RAG-FGO adopts an early-stopping strategy: when the top-scoring agent in the memory pool remains unchanged for several consecutive iterations, we regard the search as having converged to a high-quality candidate and terminate the optimization. This ensures that the overall computational cost remains manageable.
>
> **Role of the query memory pool.**
> The query memory pool is denoted as $\mathcal{M} = \{(A_i, R_i)\}$, where $A_i$ represents a candidate agent and $R_i$ is its corresponding similarity score (i.e., $\mathrm{Sim}(h(c(A_i, q)), h^{\mathrm{ref}})$). The memory pool stores high-scoring candidate agents during the optimization process and provides two key types of information for subsequent search: **(i)** by retaining high-quality candidates, it supports local exploration around their neighborhoods; and **(ii)** by preserving a diverse set of high-scoring candidates, it enables the search to maintain stability across the entire policy space. Hence, the memory pool serves as a ``behavior buffer’’ in the search process, continuously providing stable and diverse candidate references for RAG-FGO without updating any model parameters.

---

> ### Author Response · Authors · 2025-11-25
> **Response to Reviewer p9LA  [4/4]**
>
> ---
> > **Q1: What is the implementation of Agent initialization in line 198, and what is the latency/cost to perform one iteration among agents collaboration?**
>
> **Response:**
>
> **(1) What is the implementation of Agent initialization in line 198.**
>
>
> We appreciate the reviewer's attention to the agent initialization procedure described around line 198. In our formulation, each agent is defined as $A_j = (C_j, \theta_j)$. In practice, these agents are implicitly instantiated through repeated independent sampling of prompt templates and parameter vectors. During initialization, the system randomly selects an initial query cue (corresponding to $C_j$) from a predefined pool of prompt templates. This pool is implemented in the codebase through a factory class that manages the diverse prompt space described in the paper. Meanwhile, each agent's parameter vector $\theta_j$ is generated via random sampling, including generation-related hyperparameters such as temperature top-$p$, frequency\_penalty, presence\_penalty, and max\_tokens. This sampling procedure corresponds to the paper's statement that parameters are ``initialized from distributions to ensure diversity.'' The loop around line 198 repeatedly applies this sampling process, and each iteration produces a new combination $(C_j, \theta_j)$. In this way, we obtain the initial agent set $\{A_1, \dots, A_N\}$ as described in the paper.
>
> **(2) what is the latency/cost to perform one iteration among agents collaboration?**
>
> We thank the reviewer for the question. The table below reports the average token cost and runtime per iteration on the HotpotQA dataset. In addition, we provide further results on resource consumption in **Section 4.2 Performance Evaluation** of the revised manuscript.
> | mc | Cost per Iteration (tokens) | Runtime (seconds) |
> |----|------------------------------|--------------------|
> | 3  | $10,842 \pm 540$             | $6.8 \pm 1.7$      |
> | 6  | $17,231 \pm 910$             | $14.6 \pm 2.3$     |
> | 9  | $22,914 \pm 1,210$           | $20.7 \pm 3.9$     |
>
> **References：**
>
> [1] Lewis, P., Perez, E., Piktus, A., Petroni, F., Karpukhin, V., Goyal, N., ... \& Kiela, D. (2020). Retrieval-augmented generation for knowledge-intensive nlp tasks. Advances in neural information processing systems, 33, 9459-9474.
>
> [2] Izacard, G., \& Grave, E. (2021, April). Leveraging passage retrieval with generative models for open domain question answering. In Proceedings of the 16th conference of the european chapter of the association for computational linguistics: main volume (pp. 874-880).
>
> [3] Kim, K., \& Lee, J. Y. (2024). Re-rag: Improving open-domain qa performance and interpretability with relevance estimator in retrieval-augmented generation. arXiv preprint arXiv:2406.05794.
>
> ---
> **We thank you again for reviewing our work. Please let us know if we misunderstood any of your questions, or if you have any follow-up on our responses. We will be happy to provide further clarification at any time.**

---

### Meta-Review · Area_Chair_Mf1K · 2026-01-06

**Summary:**

This paper proposes RAG-FGO, a bio-inspired framework that uses the Fungal Growth Optimizer (FGO) to optimize retrieval agents (prompt templates and decoding parameters) for generative RAG. The method formulates retrieval optimization as a hybrid discrete-continuous search problem, combining global exploration and local refinement via a query memory pool. Evaluations on QA benchmarks show improvements over several agentic and traditional RAG baselines.

**Reviewer Concerns:**

The authors have diligently conducted numerous new experiments, addressing many specific requests around evaluations, ablations, and cost analysis. However, I think fundamental concerns about the paper's theoretical depth and algorithmic novelty remain unresolved.

I thank the authors have made great efforts for rebuttal, while it seems that lots of revisions to be included into the revised version. Therefore, I personally think a signficant revision of this work, with subtanlly improved quality, for another venue would be a better choice.

**Reviewer Scores:**

I think the reviewers may not signficiantly change their scores.

---

### Decision · Program_Chairs · 2026-01-26

Reject